# Selective sonochemical post-synthesis modification of LTA zeolite with zinc species

Jesús Isaías De León-Ramírez[1,2]*, Víctor Alfredo Reyes Villegas[1,2],
Sergio Pérez-Sicairos[3], José Román Chávez Méndez[4], Fernando Chávez-Rivas[5],
Rosario Isidro Yocupicio-Gaxiola[6], Vitalii Petranovskii[1]*

1 Centro de Nanociencias y Nanotecnología (CNyN) – Universidad Nacional Autónoma de México (UNAM), Ensenada, Baja California, México, 2 Centro de Investigación Científica y de Educación Superior de Ensenada, Ensenada, Baja California, México, 3 Centro de Graduados, ITT, Instituto Tecnológico De Tijuana, Tijuana, Baja California, México, 4 Facultad de Ciencias de la Salud, Valle de las palmas, Universidad Autónoma De Baja California, Tijuana, Baja California, México, 5 Departamento de Física, ESFM, Instituto Politécnico Nacional, México, Distrito Federal, Mexico, 6 Instituto Superior de Guasave, Carretera Internacional Entronque a Brecha, Guasave, Sinaloa

* Jesus.deleon@ens.cnyn.unam.mx (JIDR); vitalii@ens.cnyn.unam.mx (VP)

## Abstract

LTA zeolite is known for its straightforward synthesis and well-defined structure, composed of α and β cages. This structure offers a versatile platform for hosting molecules of varying sizes. In the sodic form (NaA), the Na$^+$ ions are located within these cages' cation exchange sites. When substituted with zinc species, a precise tuning of pore size and properties can emerge. In this study, we employed an eco-friendly post-synthesis modification of a NaA zeolite with zinc species: ZnO, ZnO$_2$, and Zn(OH)$_2$ by a sono-assisted deposition method. This approach allowed a short deposition time (30 min), providing control over the desired Zn species incorporated into NaA zeolite. Equally, the facile selective integration of zinc species in the form of supported nanoparticles and Zn$^{2+}$ in cation exchange sites modified the physico-chemical properties of the material, including unique surface charge redistribution and pore architectures. These results highlight the potential of sono-assisted deposition as an alternative strategy for engineering advanced zeolitic materials, opening new avenues for innovative applications in catalysis, adsorption, and beyond.

## Introduction

Among the zeolite family, an important place is occupied by the synthetic zeolite Linde Type A, also known as Zeolite A, or LTA, according to the three-letter abbreviations introduced by the International Zeolite Association (IZA) to distinguish between topologically different zeolite structures [1]. LTA zeolite has a structure built from double four-membered ring units (D4R) and single six-membered ring structures (S6R). These zeolitic crystal structures form two types of approximately spherical cavities called β-cage (sodalite cage, 6.6 Å in free diameter) and an α-cage (supercage, 11.4

**Data availability statement:** All relevant data are within the paper and its Supporting Information files.

**Funding:** the funders (National Autonomous University of Mexico, UNAM) had no role in study design, data collection and analysis, decision to publish, or preparation of the manuscript. Regarding funding information, we would like the following text to be presented in the "Funding Statement" section: This work was supported by the UNAM (National Autonomous University of Mexico) through time in the supercomputer from the projects LANCAD-UNAM-DGTIC-423 y 084, and the given resources from the CB-CONAHCYT A1-S-33492 y PAPIIT-DGAPA-UNAM IG101623 projects. As well as the support of the department in zeolite investigations through the academic staff (zeolite investigations) from the ICUAP (Institute of Science of the Benemerit Autonomous University of Puebla).

**Competing interests:** The authors have declared that no competing interests exist.

Å). The free diameter of the α-cage is reached through an 8-membered entrance window. This cavity diameter in zeolite A depends on the cations occupying the cation exchange sites, such as $K^+$, $Na^+$, or $Ca^{2+}$ (being KA, NaA, and CaA, respectively), which allow the adsorption of molecules with kinetic diameters less than 3, 4, and 5 angstroms (Å) hence the commercial names of 3A, 4A, and 5A for zeolites, respectively [2].

Due to its crystalline framework consisting of well-organized arrays of alternating $(SiO_4)$ and $(AlO_4^-)$ tetrahedra, LTA zeolite exhibits unique structural properties. The negative charges of substituted Al tetrahedra are compensated by cations (usually $Na^+$), which are weakly bound to the zeolite structure and are easily exchangeable [3]. This versatility enables modifications tailored to specific applications. Furthermore, LTA's secondary building units (SBUs) are double four-membered rings (D4R), contributing to its robust and versatile framework [4].

LTA zeolite's exceptional ion-exchange properties and high surface area make it widely used in applications such as water purification, gas adsorption, and catalysis [5–10]. A key focus of recent studies is the functionalization of LTA by incorporating nanoparticles to enhance its properties. Such modifications exploit the nanoporosity of zeolite, which for LTA is approximately 48%, free space that serves to host nanoparticles and prevent their agglomeration [11–13].

Researchers have explored Zn-based compounds such as ZnO [14–17], $Zn(OH)_2$ [18,19], and $ZnO_2$ [20–22] to develop advanced materials for various applications. These zinc-based nanoparticles have garnered significant attention for their catalytic, photocatalytic, and oligodynamic properties. The interaction of Zinc with biological systems further underscores its relevance. Considering that ZnO is the main compound of interest in the literature that is often stabilized and most studied within zeolite matrices (ZnO@LTA) [23–27], we propose an ultrasonic method for the selective synthesis and support of the oxide, hydroxide and peroxide forms of Zn modified LTA zeolite (ZnO@NaA, $Zn(OH)_2$@NaA and $ZnO_2$@NaA).

Ultrasound can influence various stages of the synthesis process of zeolitic materials: before, during, and after crystallization. The application of sonochemical methods has gained relevance in the fields of crystallization, due to its ability to enhance and accelerate crystal formation by reducing the induction period. These improvements are achieved through acoustic cavitation, where ultrasonic waves induce the collapse of bubbles that generate high-energy microenvironments, releasing free radicals that promote controlled nucleation and accelerate crystal growth. This approach, known as sonocrystallization, allows for a reduction in synthesis time, a decrease in average crystal size, improved particle size distribution, and better control over material morphology. Furthermore, the simplicity and efficiency of this method make it a promising alternative to conventional hydrothermal methods for obtaining zeolitic materials [28].

This ultrasonic irradiation can be used in the pre-synthesis stage, particularly during the physical mixing of different zeolite types, and has proven to be an effective method. The physical effects of ultrasound in heterogeneous media (liquid-solid) generate cavitation that leads to microbubbles' formation and collapse, producing shock waves, high temperatures, and localized high pressures. These physical effects lead

to the breakup of particle agglomerates, promoting a more uniform and stable dispersion. Several studies have reported that this technique improves mixture homogeneity, increases surface area, reduces particle size, and shortens preparation time, thereby optimizing conditions for subsequent synthesis [29].

On the other hand, ultrasound has also been employed as a post-synthesis treatment for structural modification of zeolites. Cavitation in zeolite suspensions generates additional microscopic agitation, improving local mass transfer [30]. The most commonly used post-synthesis methods for introducing metal species (conventional methods) are impregnation and co-precipitation [31], yielding the deposition of nanoparticles on zeolite surfaces. Nevertheless, ultrasonic irradiation has attracted attention due to its ability to enhance nanoparticle formation, migration, stabilization, and immobilization on the zeolitic surfaces [32]. These effects are attributed to the extreme localized environments generated by ultrasonic waves, such as elevated temperatures and pressures, which enhance: (i) nanoparticle synthesis, (ii) surface deposition, and (iii) homogenous dispersion [33].

Nevertheless, despite the multiple documented benefits of ultrasound at various stages of zeolite modification, its specific application in incorporating metal species such as Zn into NaA-type zeolites has been scarcely explored. Traditional methods for this incorporation present significant limitations, such as poor metal dispersion and the time-consuming nature of the process [34,35]. In this context, sono-assisted techniques represent a promising yet underdeveloped strategy to overcome these limitations.

This study addresses this research gap by employing a systematic sonochemical approach to modify LTA (NaA) zeolite with different Zn species. Selectively synthesizing and dispersing through the zeolitic surface zinc oxide, hydroxide and peroxide nanoparticles. The impact of ultrasonic irradiation on the Zn incorporation, zeolite morphology, and surface properties of the material Zn-modified LTA was explored. By leveraging the exceptional ion-exchange capabilities of LTA zeolite, this research contributes to the growing field of sonochemistry in zeolitic materials functionalized with metal nanoparticles, proposing potential applications in catalysis, bioengineering, and beyond.

## Experimental

### Synthesis of materials

**Synthesis of zeolite LTA.** LTA zeolite was synthesized using a hydrothermal method based on the procedure proposed by R. W. Thompson and K. C. Franklin [36], with slight precursor modifications: NaOH (Catalog No. 221465), $Al(OH)_4Na$ (Catalog No. 13404) and $Na_2O(SiO_2)x \cdot xH_2O$ (Catalog No.338443) were supplied by Sigma Aldrich. Initially, sodium hydroxide (NaOH, ≥98%, Sigma-Aldrich) was dissolved in 4 L of deionized water under gentle stirring until fully dissolved. The resulting solution was equally divided into two 2 L portions and transferred into polypropylene bottles.

One portion of sodium aluminate ($NaAlO_2$, Sigma-Aldrich), sodium aluminate ($NaAlO_2$, Sigma-Aldrich) was added and mixed in a sealed container until a clear solution was obtained. In the second portion, commercial sodium silicate solution was added in sufficient amounts to maintain an appropriate Si/Al molar ratio. The mixture was stirred until homogeneous. The sodium silicate solution was quickly poured into the aluminate solution, forming a viscous gel. The container was tightly sealed and manually shaken until a uniform gel was obtained. The gel was transferred to a polypropylene bottle and subjected to hydrothermal crystallization treatment at 99 ± 1 °C for 4 hours under static conditions.

After crystallization, the suspensions were allowed to cool to room temperature (<30 °C), and the solid product was recovered by decantation. The solids were washed with excess deionized water until the supernatant reached a pH < 9.

Finally, the material was dried by separating on different filter papers and placed on a watch glass at 80°C for 24 hours. The final yield was approximately 500 g of dry material, and the resulting zeolite powder in the sodium form was designated as NaA.

**Deposition of zinc species.** The speciation diagrams were modeled using the Hydra/Medusa software, illustrating the fraction of zinc species at different pH values and concentrations (S1 Fig). Based on these diagrams, the concentration and pH conditions were carefully adjusted to favor the desired zinc species' selective deposition by promoting a

specific species' predominance under controlled conditions. It was shown that $Zn(OH)_2$ formation is favored at low zinc concentrations. In contrast, ZnO becomes the predominant species at higher concentrations and under alkaline conditions. Hence, this specific predominance promoted the desired zinc species' selective deposition.

In brief, NaA was suspended in water, and by a sono-assisted method, $Zn(OH)_2$, ZnO, or $ZnO_2$ were synthesized and deposited on the zeolitic surface. Initially, 1 g of NaA was dispersed in 25 mL of water using ultrasonic irradiation for 5 min and preheated to 65 ℃ (step done in triplicate, for each Zn species). For the precipitation of $Zn(OH)_2$, 10 mL of 0.05 mM solution of zinc acetate $(Zn(CH_3COOH)_2)$ was added dropwise to the 25 mL zeolitic suspension, adjusting to a final pH value of 12, with concentrated $NH_4OH$ [37]. For the deposition of ZnO species, 25 mL of 0.2 mM aqueous solution of $Zn(CH_3COOH)_2$ was prepared and added to a 25 mL zeolitic suspension, carefully adjusting the final pH value to 8, employing acetic acid or $NH_4OH$ when necessary [33]. For $ZnO_2$, 1 g of zinc acetate was dispersed in 20 mL of distilled water, preheated to 65 ℃, and mixed under ultrasound for 2 min with 5 mL of 3% hydrogen peroxide [38], and added dropwise to a 25 mL NaA suspension. The final pH for all the prepared suspensions was adjusted during the dropwise addition of the Zn precursor solutions with concentrated HCl or $NH_4OH$. Once the desired pH was reached, $H_2O$ was added to the 50 mL mark and allowed to react under ultrasonic irradiation for 30 min at 65 ℃. The well-formed precipitates were filtered under vacuum through a 0.45 μm membrane and washed with excess distilled water until a neutral value (pH = 7) was achieved. The products were dried at 35 ℃ for 24 hours and labeled $Zn(OH)_2$@NaA, ZnO@NaA, and $ZnO_2$@NaA [39].

To evaluate the effect of the sonication conditions on the zeolite structure, a NaA suspension was sonicated at the deposition conditions (pH, ultrasound duration, and temperature), with no addition of $Zn^{2+}$ precursor (Blank samples). These blank samples, (OH)@NaA and $(O_2)$@NaA, were analyzed by X-ray diffraction analysis, and given that no difference was observed, these samples were excluded from the study. This data is presented as support information (S2 Fig).

## Characterization

The morphology of all samples was obtained from scanning electron microscopy (SEM) measurements (JEOL JIB-4500 equipment, MA, USA); the same equipment was used to obtain the chemical analysis data by X-ray dispersive energy spectroscopy (EDX). These data were compared with transmission electron microscopy (TEM, Hitachi H7500, Hitachi Ltd., Tokyo, Japan) at 80 kV. Calculating the particle size distribution by treating the micrographs using the ImageJ software.

The chemical composition was evaluated by inductively coupled plasma-optical emission spectroscopy (ICP-OES) using a Vista-MPX CCD (CO, USA) simultaneously with ICP-OES (Varian) (CO, USA). A quantity between 25–50 mg of each sample (NaA, $Zn(OH)_2$@NaA, ZnO@NaA, or $ZnO_2$@NaA) was dissolved by adding 2 mL of $HNO_3$ and 1 mL of concentrated HF and left for 24 h at 40 ℃. Subsequently, 40 mL of 2% $H_3BO_3$ was added and replaced for 5 h at 40 ℃. Before the measurements, all samples were diluted by a factor of 25. The atomic weight percentages were obtained by calculating the number of moles of each element divided by the total moles of all elements in the formula.

X-ray diffraction (XRD) analysis was performed in an Aeris XRD diffractometer from Malvern Panalytical using a wavelength of λ = 0.154056 nm, with a voltage of 40 kV and a current of 15 mA in the Cu Kα line. The resulting diffraction patterns were processed using the X'Pert HighScore Plus software, performing a Rietveld refinement analysis. The refinement was used to calculate the phase quantities present, and the corresponding phases were obtained from the Crystallography Open Database (COD): NaA 7117433, ZnA 1541696, $Zn(OH)_2$ 1011223, ZnO 1011258, and $ZnO_2$ 1762857.

To study the binding vibrations of the Zn species and the zeolite, Fourier transform infrared spectroscopy (FT-IR) and Raman spectroscopy were employed. The spectra were obtained by a Nicolet IS10 Thermo Scientific FT-IR equipment (Waltham, MA, USA), with the ATR attachment loading approximately 100 mg of powder sample. The method used was a sweep of 500 cm$^{-1}$ to 4000 cm$^{-1}$ (100 scans were performed per reading, and 3 readings were carried out). Raman spectra were collected using a LabRAM HR 800 confocal micro-Raman spectrometer (Horiba Jobin-Yvon, HORIBA Instruments Inc., Piscataway, NJ, USA), equipped with an OLYMPUS BX41 microscope. The samples were placed on the microscope

stage and examined at room temperature. A 633 nm He-Ne laser was employed as the excitation source, with a nominal output power of 17 mW, attenuated using a neutral density filter (D3, I/1000) to prevent thermal damage to the sample. Spectra were acquired from at least three points on each sample to ensure reproducibility. A 100 × objective lens focused the laser beam on the material's surface. The integration time was 8 seconds, and the spectra were collected in the 100–1500 cm$^{-1}$ range.

$N_2$ adsorption-desorption isotherm measurements were used to determine the textural properties, surface area ($S_a$), pore volume ($V_p$), and pore diameter ($D_P$) of the studied materials at −196 °C with $P/P_0$ values of 0–0.99 in a Quantachrome Instruments autosorb iQ device. The studied materials were degassed at 50°C for 60 minutes, followed by a heating ramp with a rate of 5°C min$^{-1}$ until reaching 250°C, maintaining this temperature for 360 minutes. The data was processed with the Quantachrome ASiQwin software using the Langmuir equation to calculate the specific surface areas ($S_a$) and NLDFT for the cumulative pore volume ($V_p$) from the saturation point ($P/P_0 \sim 0.7$). For micropore area and volume ($S_{micro}$ and $V_{micro}$), a *t-plot* was used along with the NLDFT data to model the micropore size distribution and calculate the mean pore diameter ($D_P$). The mesopore area ($S_{meso} = S_a - S_{micro}$) and volume ($V_{meso} = V_p - V_{micro}$) were obtained from the differences between the total and micropore values. The obtained models and standard deviations are presented as supporting information (S1 Table).

The thermal stability of the prepared zeolites was evaluated using TGA analysis, performed with a TA instrument, model Q600. The study was conducted in an $O_2$ atmosphere from 30 °C to 800 °C, with a heating ramp at a rate of 20 °C min$^{-1}$. To determine the zeta potential ($\zeta$), the samples were resuspended in water with the help of an ultrasonic bath at room temperature (>5 min) until a stable suspension was reached. The $\zeta$ measurements were then performed in Zetasizer Nano ZS (from Malvern Instruments) by averaging three automatic readings at 25 ºC, with no pause between measurements. Diffuse reflectance spectroscopy (DRS) measurements in the ultraviolet-visible (UV-Vis) range were performed using a Cary 5000 UV-Vis NIR spectrophotometer (Richmond, CA, USA) in a wavelength range of 200–1200 nm at room temperature. The band gap ($E_g$) was calculated from the obtained spectra by applying the Kubelka-Munk function [40].

All spectra (FT-IR, Raman and UV-Vis DRS) were processed and analyzed using OriginPro 2023 (OriginLab Corporation), including baseline correction and peak deconvolution where applicable.

## Results and discussion

### Particle size, morphology, and composition

The pristine NaA zeolite has a well-defined cubic crystal structure (average size of 2 μm) characterized by SEM (Fig 1A). When LTA is crystallized in a cubic structure, the morphology is correlated with a Si/Al ratio of 1.0, due to the selective formation of the narrow compositional range of zeolite A [41,42]. After the treatments the morphology of zeolite A remains in a cubic shape (Fig 1), indicating that the Si/Al ratio is maintained, supported by the chemical composition analysis.

After the treatment to deposit $Zn(OH)_2$ species (Fig 1B) a slight change in the NaA structure was detected. Change attributed to the low quantity of zinc precursor used in the sono-assisted deposition process (Table 1). Suggesting that the formation of these particles is minimal or even null, despite undergoing more extreme pH conditions and identical ultrasonic and thermal treatments. On the other hand, when depositing ZnO nanoparticles (ZnO@NaA sample, Fig 1C), the NaA particle is shown with a dispersed material on the surface. Suggesting the formation of ZnO nanoparticles supported on the NaA surface [43]. Interestingly, this was the only sample where nanoparticles were observed on the outer surface of the zeolitic structure and where zinc was detected via the EDS analysis (Table 1). Equally, both ZnO@NaA and $ZnO_2$@NaA samples (Fig 1C and 1D) show subtle changes in the edges and morphology of NaA. The zeolite is surrounded by clusters of nanoparticles after the ZnO deposition treatment and with less clusters of the $ZnO_2$ nanoparticles deposition treatment. Nevertheless, the overall cubic shape morphology is maintained. Attributing these changes to ions penetrating the zeolite structure, altering its roughness [44]. Concluding that these external clusters result from the introduction of defects in the zeolite structure by the combination of sonochemical treatments and the ions in solution.

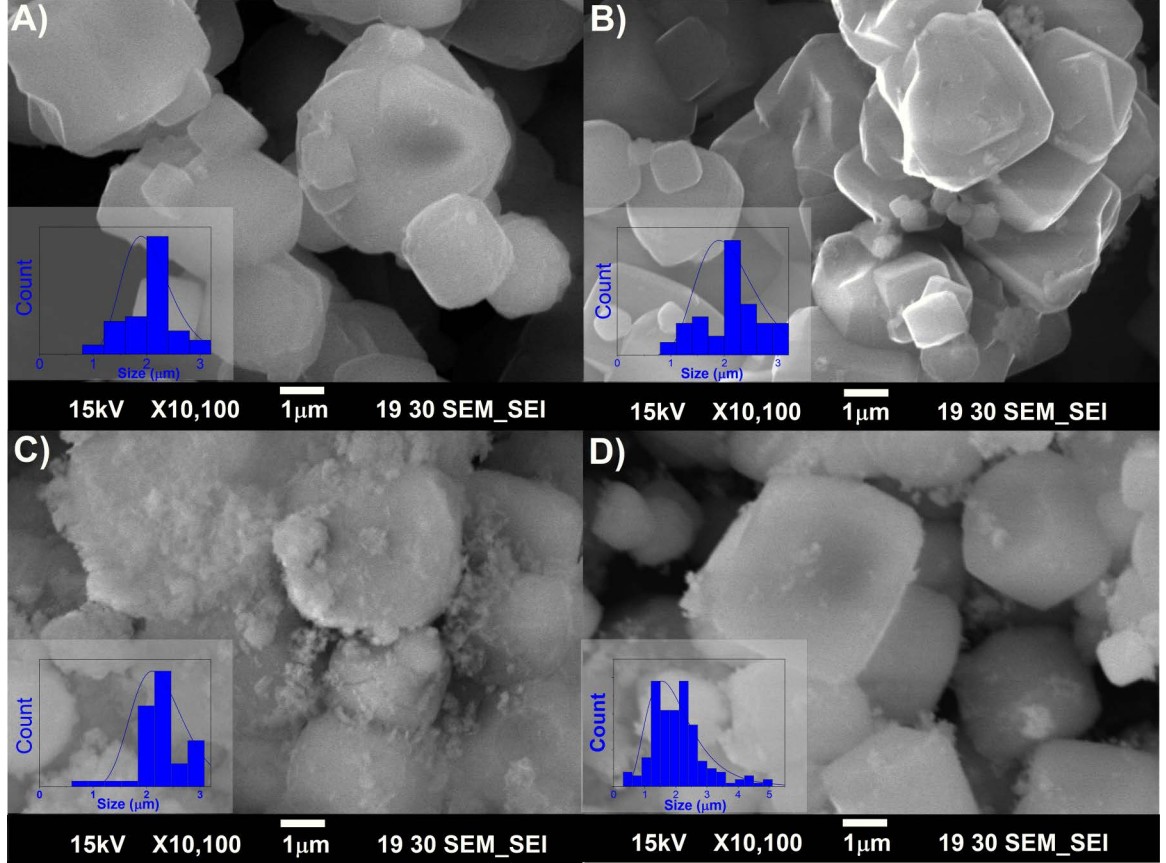

**Fig 1. Scanning electron microscopy (SEM).** A) Micrographs of cubic NaA zeolite particles, untreated and after ultrasonic treatment, under B) Zn(OH)2, C) ZnO, and D) ZnO$_2$ nanoparticle synthesis and deposition conditions. Insets in all micrographs correspond to their particle size distribution.

**Table 1. Chemical analysis from inductively coupled plasma optical emission spectroscopy (ICP-OES) and energy-dispersive X-ray spectroscopy (EDS).**

| Sample | Technique | Atomic % | | | | | | Molar Ratios |
|---|---|---|---|---|---|---|---|---|
| | | Na | Si | Al | O | Zn | Chemical Formula | (Na:Al:Si:Zn) |
| NaA | EDS | 11 | 11 | 11 | 66 | 0 | $|Na_{12}|[Al_{12}Si_{12}O_{48}]$ | 1:1:1:0 |
| | ICP-OES | 34 | 33 | 33 | – | 0 | $|Na_{12}|[Al_{12}Si_{12}]O$ | 1:1:1:0 |
| Zn(OH)$_2$@NaA | EDS | 13 | 12 | 12 | 63 | 0 | $|Na_{12}|[Al_{12}Si_{12}O_{48}]$ | 1:1:1:0 |
| | ICP-OES | 33 | 33 | 33 | – | 1 | $|Na_{11}|[Al_{12}Si_{12}O_{48}]@Zn_{0.18}O$ | 1:1:1:1 |
| ZnO@NaA | EDS | 3 | 11 | 11 | 62 | 13 | $|Na_6Zn_3|[Al_{12}Si_{12}O_{48}]@Zn_{13}O_{13}$ | 2:1:1:2 |
| | ICP-OES | 11 | 33 | 33 | – | 23 | $|Na_4Zn_4|[Al_{12}Si_{12}O\_]@Zn_5O$ | 1:1:1:2 |
| ZnO$_2$@NaA | EDS | 5 | 11 | 11 | 68 | 5 | $|Na_6Zn_3|[Al_{12}Si_{12}O_{48}]@Zn_3O_6$ | 2:1:1:1 |
| | ICP-OES | 16 | 34 | 34 | – | 16 | $|Na_6Zn_3|[Al_{12}Si_{12}O\_]@Zn_3O$ | 1:1:1:2 |

All the values were rounded to the minimum significant digits for convenience and simplification.

Furthermore, after applying ultrasound in the presence of the precursors for Zn nanoparticle synthesis, the smooth surface of the NaA zeolite is modified (Fig 2). This change suggests the presence of particles on the eroded surface distributed throughout the zeolite [45]. This phenomenon can be explained by the stress and erosion caused to the zeolitic

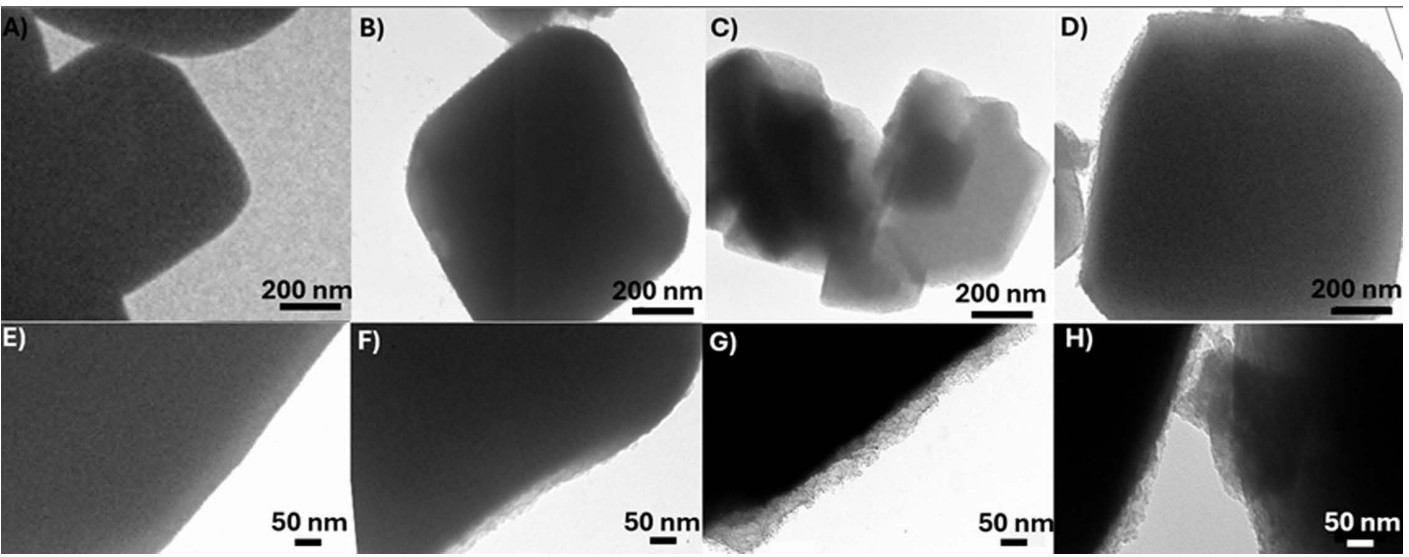

**Fig 2. Transmission electron microscopy (TEM).** A) Micrographs of NaA, B) Zn(OH)$_2$@NaA, C) ZnO@NaA, and D) ZnO$_2$@NaA, and their corresponding edges E) NaA, F) Zn(OH)$_2$@NaA, G) ZnO@NaA, and H) ZnO$_2$@NaA.

surface when interacting with the cavitation and microjet effects caused by ultrasonic waves [46]. Hence, defects are added to the samples and serve as nucleation sites, guiding the nanoparticle crystallization to these eroded surface sites.

Comparing the particle size distributions obtained from SEM and TEM, similarities are observed in contrast to XRD. The detected variation from the Scherrer equation suggests that the NaA zeolite samples are 80 nm crystallites forming 2 µm ± 0.3 polycrystalline particles.

Although the typical composition of NaA (Si/Al ratio of 1) is maintained, Zn was introduced and detected in all the Zn treated samples (Table 1). However, the deposition of Zn(OH)$_2$ had no significant change when measured by EDS, with a minimal amount of zinc quantified from the ICP technique. Meaning that only ICP was sensitive enough to detect the low concentration of the deposited Zn(OH)$_2$. A higher concentration of this Zn species is needed for detection by other less sensitive methods. This indicates that the chosen conditions for depositing Zn(OH)$_2$ species only allow a small amount of zinc hydroxide to precipitate onto the NaA structure.

As for the ZnO@NaA, a significant amount of zinc was detected in exchanged form, and an additional portion in the form of oxide. The exchanged sodium decreased to half, indicating that half of the cation exchange sites are now balanced with Zn$^{2+}$. When calculating the molar ratio for Zn in this sample (Table 1, EDS), the close 1:1 ratio in Zn:O agrees with the theoretical molar ratio of ZnO species. Similarly, the ZnO$_2$@NaA sample exhibited Zn$^{2+}$ in cation exchange sites coexisting with the ZnO$_2$ species. This co-existing phenomenon was attributed to ultrasound, favoring the synthesis of species on zeolite and providing enhanced ion exchange conditions.

## Textural properties

The deposition of the detected zinc nanoparticles causes changes in the adsorption properties of the zeolite surface (Fig 3). According to the IUPAC classification, the NaA sample displays a type I microporous isotherm [47]. The adsorption-desorption profile of the Zn(OH)$_2$@NaA sample was characteristic of an H1 loop, suggesting a pore network with an "ink-bottle" structure, where the width of the neck size distribution is like the width of the pore or cavity size distribution [47,48]. This phenomenon may be caused by the deposition of Zn species in the internal zeolitic surface, modifying the characteristic pore architecture and resulting in a narrow loop (Fig 3).

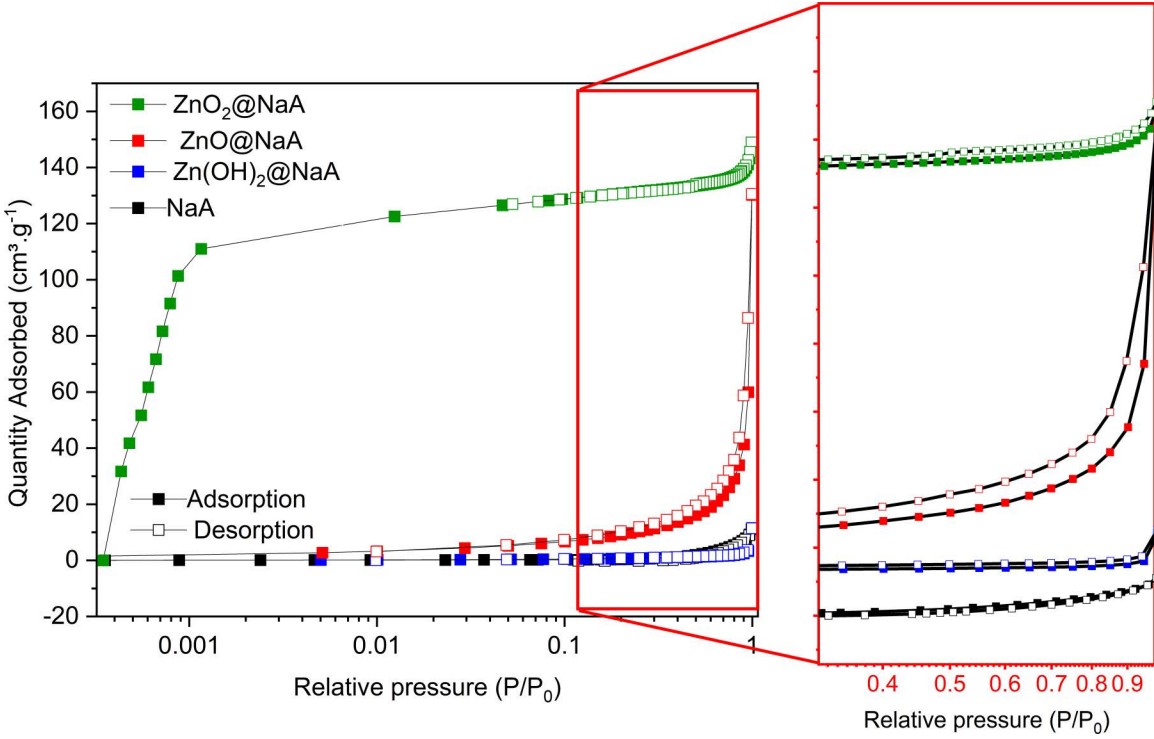

**Fig 3. N$_2$ adsorption-desorption isotherms.** Zeolite NaA (black) modified with Zn(OH)$_2$@NaA (blue line), ZnO@NaA (red line), ZnO$_2$@NaA (green line).

Similarly, the deposition of ZnO also causes a similar H1 isotherm change with "ink-bottle" structure pores [47,48]. Suggesting that ZnO nanoparticles are located within the NaA pore network. These changes in particle distributions inside the pores lead to the observed variations in the isotherms. In addition, the increase in surface area at the P/P$_0$ < 0.9 is attributed to a rise in intercrystalline voids and the broader loop of ZnO compared to Zn(OH)$_2$, which hints at a decrease in pore uniformity and connectivity [49]. Conversely, ZnO$_2$ exhibits a type IV H4 loop (Fig 3), with a pronounced uptake at low p/p$_0$ ratios associated with micropore filling. H4 loops are often found in aggregated zeolite crystals or nanoaggregates, where the presence of ZnO$_2$ may cause. Furthermore, this loop type has been observed in zeolites supporting nanoparticles in small surface crevices [50].

It is worth mentioning that the isotherm of NaA zeolite (Fig 3) shows a negative trend in gas adsorption; this may be due to the pores of the zeolite being blocked, preventing the correct adsorption of the gas. To rule out this hypothesis, a washing process using ultrasound and temperature was carried out to eliminate impurities in the zeolite pores. However, this trend remained negative, as observed in the supporting information (S3 Fig).

The formula considered in these isotherms, by correcting the non-ideality of the gas, is shown in equation 1 (Eq. 1):

Eq. 1: The non-ideality correction is replaced by P with P(1 + αP), where α is the non-ideality factor.

This gives:

$$n = \frac{PV}{RT}(1 + \alpha P) \tag{1}$$

where:

• $n$ is the number of moles of the gas,

- P is the pressure,

- V is the volume,

- R is the ideal gas constant,

- T is the temperature.

Subsequently, a correction was made, considering the compressibility of the gas (Eq. 2) from the real gas equation, where the isotherm improves (Fig 5, black line).

$$n = \frac{PV}{RT_Z(P, T)}$$

(2)

Where Z(P, T) is the compressibility factor for the gas at a given pressure and temperature.

This procedure yielded results consistent with the previously observed negative adsorption. In some instances, an unexpected phenomenon was observed during adsorption, where the structure underwent sudden contraction as it became saturated with gas, causing deformation [51,52]. This behavior is attributed to the pressure exerted by the gas, in this case nitrogen, which contracts the structure and reduces the pore size. This phenomenon has been studied on cubic structures and is related to the nature of LTA zeolite, known for its remarkable flexibility [48,53]. Theoretical calculations have shown that the zeolite structure can contract under pressure or temperature without causing destruction [53].

Interestingly, the measured surface area and pore volume of NaA increased when the Zn species were supported by the ultrasound-assisted method (Table 2). Finding that the Zn species must be located within the pore channels, thus increasing the rigidity of the structure and preventing its shrinkage (Fig 5). This increased stiffness allowed for more gas adsorption, hence the increase in surface area [48], facilitating gas adsorption and desorption isotherm (Fig 5). As for $Zn(OH)_2$, no significant changes were observed due to the low quantities of zinc ions available during the sono-assisted modification. However, these ions were sufficient to strain the network and prevent the generation of the detected negative area in the pristine zeolite. Equally, it was observed that the mesopore volumes of NaA samples containing zinc species were significantly larger than those of pristine NaA. This phenomenon is attributed to the combination of the stacked nanoparticles preventing the shrinkage of the structure and the gas filling of the intercrystalline voids [54].

## Interaction of Zn species with LTA

Even though these Zn species-zeolite interactions are present, the crystalline structure of NaA zeolite (COD): NaA 7117433) is not degraded (Fig 4). Equally, the diffraction patterns tend to be similar, with slight differences observed in the respective samples [55]. The refinements exhibit the presence of the NaA phase exchanged with Zn and $Zn(OH)_2$, ZnO,

**Table 2. Textural properties measured from the nitrogen adsorption-desorption isotherms.**

| Sample | $S_a$ | $S_{meso}$ | $S_{micro}$ | $V_P$ | $V_{meso}$ | $V_{micro}$ | $D_p$ |
| --- | --- | --- | --- | --- | --- | --- | --- |
| | $m^2.g^{-1}$ | | | $cm^3.g^{-1}$ | | | nm |
| NaA | 1.26±0.13 | | | 0.017 | | | 1.09 |
| $Zn(OH)_2$@NaA | 2.75±0.06 | | | 0.010 | | | 1.28 |
| ZnO@NaA | 437.88±3.48 | 79.40 | 358.48 | 0.267 | 0.157 | 0.110 | 0.91 |
| $ZnO_2$@NaA | 581.42±0.86 | 11.31 | 570.11 | 0.224 | 0.025 | 0.199 | 0.91 |

All the values were rounded to the minimum significant digits for convenience and simplification. $S_a$ = surface area (Langmuir), $S_{meso}$ = mespore surface area(*t-plot*), $S_{micro}$ = micropore surface area(*t-plot*), $V_p$ = pore volume(NLDFT), $V_{meso}$ = mesopore volume(*t-plot*), $V_{micro}$ = micropore volume(*t-plot*), $D_p$ = pore diameter (NLDFT).

and $ZnO_2$ species in percentages of 0.7%, 4.2%, and 5.6%, respectively, confirming the presence of these species in the structure. However, based on the XRD data and the low concentration of $Zn(OH)_2$, there is not enough evidence to attribute the signals of the $Zn(OH)_2$ to those appearing in the spectrum when compared to the JCPDS cards of the $(ZnOH)_2$ phase (00-038-0356). This supports the above-mentioned suggestions regarding the concentration, size, and dispersion of the precipitated species being unfavorable to be detected by the technique [56].

This NaA zeolite with a cubic structure (Si/Al = 1) features three distinct cation exchange sites. Site I, located at the center of the six sodalite cage openings, contains eight sodium cations. Site II, in the eight-membered aperture, has three sodium cations, while Site III, near the four-membered ring, has a single sodium cation. These sites are key for identifying particle distribution in NaA zeolite, with planes (220) and (400) being the most sensitive to these sites [57]. Evidencing the interaction of the Zn species with the zeolitic structure or on the surface. The changes in the diffractograms associated with the interactions between the nanoparticles and the zeolitic matrix consist of intensities of the crystallographic planes of the samples (Fig 4B). A decrease in intensity is observed in the (220), suggesting the presence of Zn ions at exchange site I in the NaA zeolite (Fig 4C). On the other hand, the patterns show an increase in the reflection

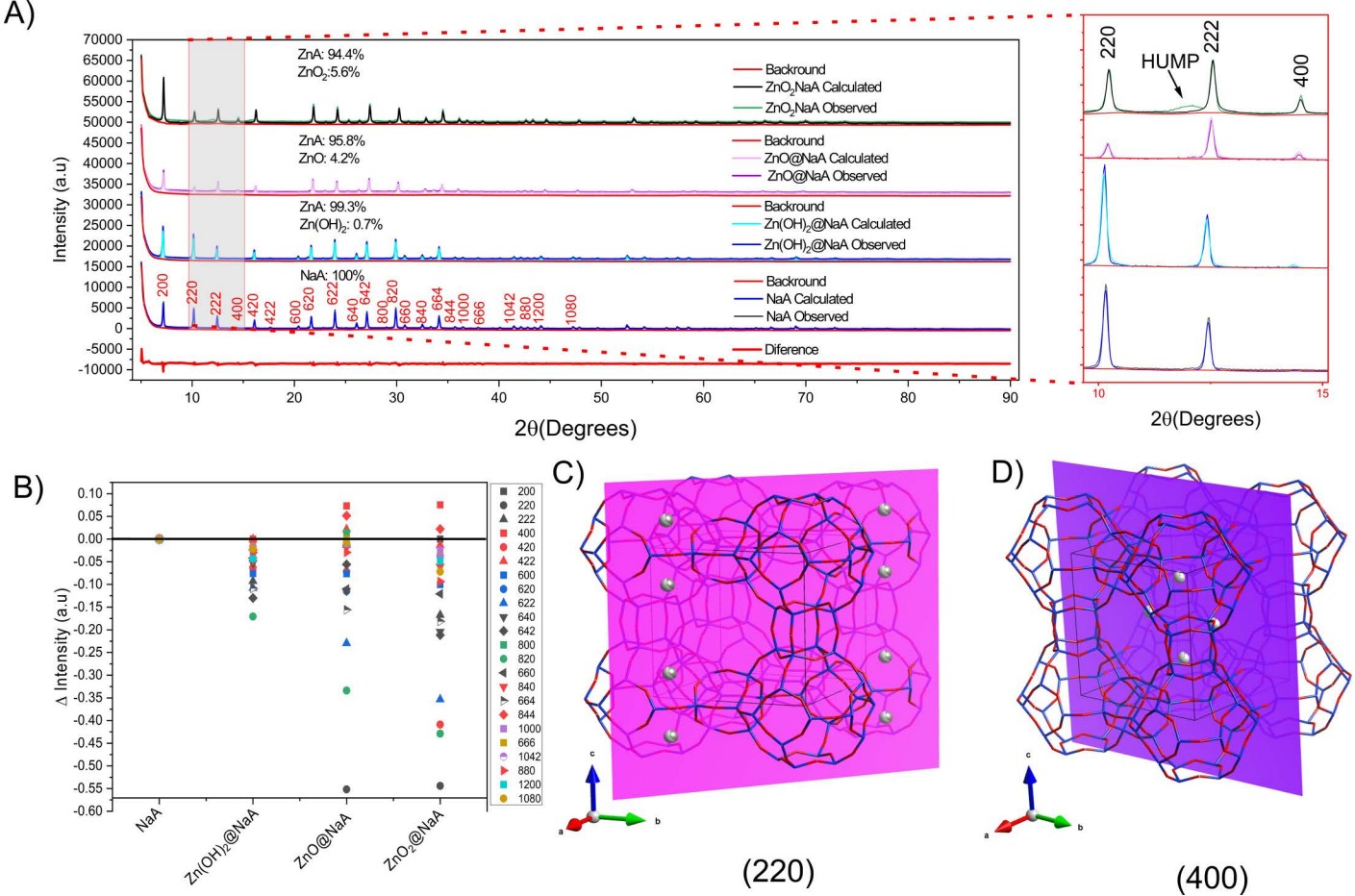

**Fig 4. X-ray diffraction measurements and analysis.** A) Difractograms of zeolite NaA modified with $Zn(OH)_2$, ZnO, and $ZnO_2$, and comparison of the observed and simulated intensities from Rietveld refinement indicating the percentage of each Zn phase and zeolite content in the zeolite. B) Differences in the normalized intensity of diffraction peaks between the planes, compared to unmodified NaA and Zn-modified NaA. C) Structural model of NaA showing a cross-section through the (220) plane and D) (440) plane in the NaA structure.

of the (400), indicating the possible incorporation of Zn within the α-cage of the zeolite (Fig 4D). Confirming that these changes in the reflection of specific planes are attributed to the interaction of foreign species (ZnO, $ZnO_2$, or $Zn(OH)_2$) with the zeolite structure [58].

In general, the detected interaction by XRD showed that Zn species interact with the zeolite crystal lattice, evidenced by a significant decrease ($p < 0.05$) in the intensity of plane (220), an increase in plane (400), and angular shifts in diffractograms. These shifts show a slight reduction in the unit cell parameters, with lattice constants decreasing from 24.6 Å in pristine NaA and $Zn(OH)_2$@NaA to 24.4 Å in ZnO@NaA and $ZnO_2$@NaA, indicating lattice contraction likely due to partial incorporation of Zn species.

In addition to the changes in the intensities of the diffractograms, shifts in the diffraction angles are observed. In the $Zn(OH)_2$@NaA sample (Table 3), a leftward shift to lower angles is observed in all planes, which may suggest that the lattice is expanding due to the incorporation of particles that are causing a movement in the crystal lattice, generating stress in the zeolite crystals. On the other hand, in the ZnO@NaA and $ZnO_2$@NaA samples, displacements in the opposite direction are observed; however, these displacements are not uniform in all planes, suggesting that the crystal lattice is undergoing distortion upon incorporation of the nanoparticles [25,59,60]. In the $ZnO_2$@NaA sample, a hump is detected on the left side of the (222) plane reflection, a phenomenon observed in samples where the lattice is distorted [61], caused by the interaction of the zeolite structure with the $ZnO_2$ species. This suggests that the interaction of $ZnO_2$ with zeolite differs from that of ZnO or $Zn(OH)_2$ species [25].

The observed XRD pattern alterations are primarily due to distortions of the structural factors caused by the integration of nanoparticles into the zeolite crystal lattice. The incorporated species causes stresses and modifications in the crystal lattice, manifested by changes in the positions of the diffraction peaks. Raman spectroscopy can corroborate this phenomenon, where a shift of the vibrational signals towards lower frequencies is observed. This shift suggests that the crystal lattice is undergoing significant distortions due to the insertion of nanoparticles, which alter the normal lattice vibrations and structural distortions [62].

From the FTIR/ATR spectroscopy (Fig 5A), the characteristic peaks of zeolite NaA [63,64]:

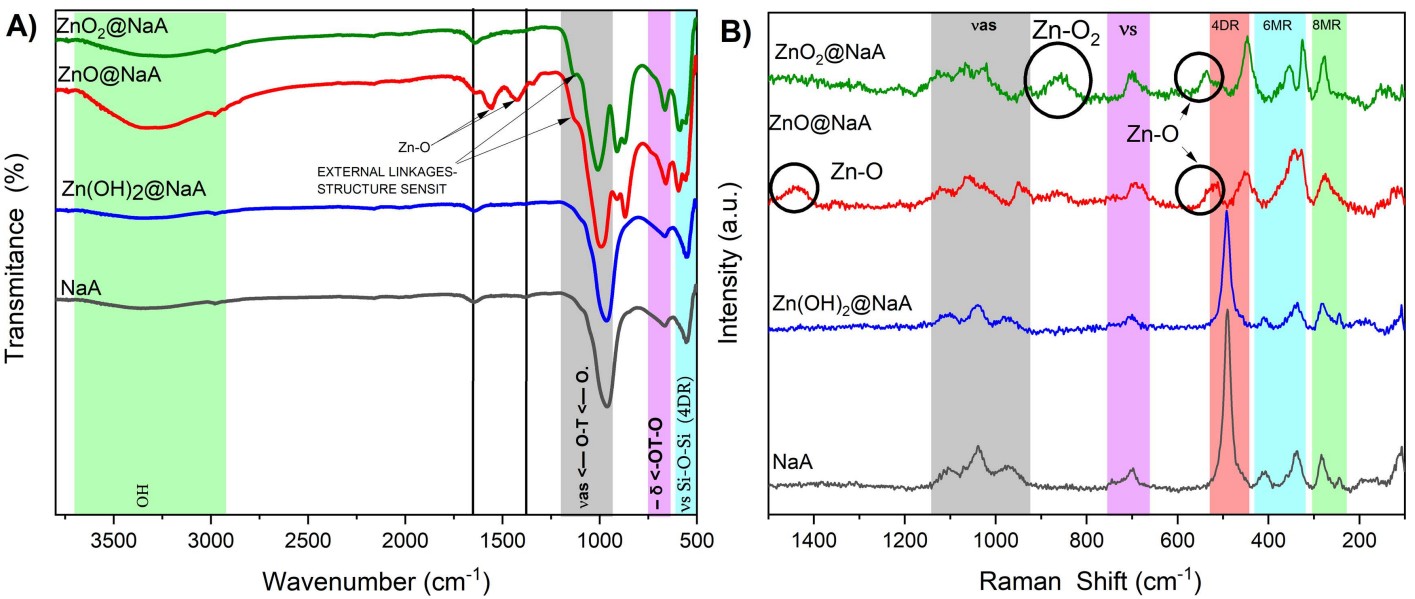

**Fig 5. Analysis of FT-IR and Raman spectra.** Zeolite NaA A) modified with $Zn(OH)_2$ (blue line), ZnO (red line), and $ZnO_2$ (green line). B) Raman vibration spectra of zeolite B) modified with $Zn(OH)_2$ (blue line), ZnO (red line), and $ZnO_2$ (green line).

**Table 3. Displacements of planes relative to NaA, where negative values indicate leftward shifts and positive values indicate rightward shifts.**

| Plane | Peak Position (2θ) | peak position shift (Δ2θ)* | | |
|---|---|---|---|---|
| | NaA | Zn(OH)$_2$@NaA | ZnO@NaA | ZnO$_2$@NaA |
| (200) | 7.18 | −0.04 | 0.03 | 0.04 |
| (220) | 10.16 | −0.02 | 0.04 | 0.06 |
| (222) | 12.44 | −0.02 | 0.09 | 0.11 |
| (420) | 16.09 | −0.02 | 0.11 | 0.15 |
| (620) | 21.65 | −0.02 | 0.15 | 0.21 |
| (622) | 23.96 | −0.02 | 0.20 | 0.25 |
| (642) | 21.66 | −0.04 | 0.15 | 0.19 |
| a=b=c (Å) | 24.6 | 24.6 | 24.4 | 24.4 |
| R$_{wp}$ | 8.5 | 12.8 | 8.1 | 13.1 |

R$_{wp}$=R$_p$- R$_{expected}$   *The values represent the peak shift difference of the samples (Δ2θ=Sample-NaA).

- 1005 cm$^{-1}$: asymmetric stretching vibrations ($v_{as}$) of Si-O(Al)

- 666 cm$^{-1}$: symmetric stretching vibrations ($v_s$) of Si-O-Al

- 554 cm$^{-1}$: (complex band) symmetric stretching vibrations ($v_s$) of Si-O-Si and bending vibrations ($v_\delta$) O-Si-O

Bands from the -OH moieties in structural water molecules and additional functional -OH groups, such as [Si(OH)Al], are usually found at about 2800–3600 cm$^{-1}$ in zeolite NaA. These bands do not show changes in the NaA sample (black line, Fig 4A). However, when NaA is subjected to the Zn modification treatments, these bands increase significantly, which may be related to the increased amount of Zn since Zn ions could be adsorbed on hydroxyls, forming bridges on the ≡Si-OH -Al≡ surface or on ≡Si-OH surface hydroxyls, ≡Al-OH hydroxyl complexes, as well as zinc hydroxides [65]. However, other authors have reported the integration of ZnO species on the surface of various zeolites, noting interactions with silanol groups and a subtle shift in the bands in these regions [54], as observed in Zn species@NaA samples. The bands around 1644 and 1322 cm$^{-1}$ are attributed to the bending of the -OH group of adsorbed water [66].

A signal shift corresponding to $v_{as}$ is observed. This may be due to the weakening of the oxygen bond or an ion disrupting this region, as can be seen in nanoparticle deposition processes [25,67,68]. This phenomenon is also sensitive to changes in the Si/Al ratio; however, this can be ruled out as chemical composition analyses (Table 2) confirm that this ratio remains constant [69].

Characteristic Zn-O bond signals are observed in ZnO@NaA near 1555 cm$^{-1}$ and 1428 cm$^{-1}$ [70]. In ZnO@NaA and ZnO$_2$@NaA, a hump is observed at 1140 cm$^{-1}$, which corresponds to a characteristic signal in zeolites of external linkages structure sensitive where ZnO and ZnO$_2$ species may interact with the NaA structure and not necessarily inside the cavities since we can correlate with the Raman spectrum [64].

ZnO@NaA and ZnO$_2$@NaA, a noticeable band around 900 cm$^{-1}$, is observed, which has also been reported to be related to the Si-O and Si-O-Al vibrations within the zeolite structure [64,66]. This band does not show a significant change in samples with a very low amount of zinc (Zn(OH)$_2$@NaA) compared to NaA. This band is associated with vibrations related to zinc; another characteristic band of ZnO nanoparticles observed is the split band 860 and 915 cm$^{-1}$, the latter being less intense [71], in contrast to ZnO$_2$, where the latter appears more intense [38]. Characteristic bands for ZnO appear near 500 cm$^{-1}$ [72], where it is observed that the 4DR vibrations are modified due to the zinc that may be incorporated in the ion exchange site. Vibrations around 500 cm$^{-1}$ show double ring vibrations in the crystal structure [69]; the shift of the bands to shorter wavelengths and the decrease in intensity may be due to the formation of new bonds with Si-O and Al-OH, and for zinc oxide and zinc peroxide a shift of the band assigned to the double rings was observed, which may demonstrate the incorporation of Zn species into the crystal structure [73].

The Raman spectra (Fig 5B) of zeolites before and after treatments reveal changes in the positions and intensity of the bands, which can vary significantly depending on the changes applied to the zeolite building units. Signal intensities are affected by particle interactions or are related to the silica-to-aluminum ratio, which can be altered by post-synthesis processes [74]. These alterations, however, are not reflected in the results of chemical composition analyses (Table 1), suggesting that they are caused by particle interactions.

The vibrational modes present in the Raman spectra of a zeolite are fundamental to identifying the secondary building units that make up the crystal. In the NaA zeolite, a breathing vibrational mode is associated with the 4-membered double ring, observable in the band at $489\,cm^{-1}$. Fig 4 shows a characteristic NaA spectrum (black line) in which this band is evident, confirming the typical NaA zeolite structure [74]. When compared with the spectra of the treated zeolites, especially the ZnO and $ZnO_2$ species, two main changes are observed: a decrease in intensity and a shift in the position of this band. These changes indicate an increased rigidity of the lattice, suggesting the integration of particles in this part of the zeolite. Furthermore, previous Raman studies of zeolite structures have shown similar changes after ion exchange with different metals, i.e., an induced zeolite framework distortion by nanoparticle interaction. As previously mentioned, the shift in the vibration of the four-membered ring is caused by the distortion of the NaA crystal lattice [62]. Correlating with the XRD results (Fig 4A), with a decrease in the intensities of the (220) passing through the double four-membered ring (Fig 4C), indicating the positioning of ions exchangeable with sodium [75]. Additionally, the shift in some angles and the appearance of a hump indicate the distortion of the lattice, which may be caused by the integration of these nanoparticles [61].

Theoretical and experimental studies indicate that Raman shifts in NaA zeolite may be due to species interactions with the LTA zeolite building units [76]. ZnO@NaA and $ZnO_2$@NaA, a very noticeable shift can be observed, which, according to DFT calculations, may be attributed to the interaction with 4DR by the species, causing individual distortion of 4DR. On the other hand, the appearance of the bands in ZnO [77,78] and $ZnO_2$ [38] confirms that the Zn species are interacting with DR4 and generating this distortion. In contrast, $Zn(OH)_2$@Na species are observed, these interactions can be verified by the appearance of FTIR bands around $560\,cm^{-1}$. Given that $Zn(OH)_2$@NaA sample has no change in the building unit, no interaction is suggested in these structures [79–81]. In the spectra of $ZnO_2$@NaA and ZnO@NaA, we can observe the appearance of a pore-opening band [81] in agreement with the BET analysis (Table 2).

NaA exhibits characteristic vibrations for the 8-membered ring (8MR) at $283\,cm^{-1}$ and the 6-membered ring (6MR) at $338\,cm^{-1}$ and $410\,cm^{-1}$. Asymmetric vibrations appear at $700-704\,cm^{-1}$, $971-977\,cm^{-1}$, $1040\,cm^{-1}$, and $1100-1106\,cm^{-1}$ [82]. For the 8MR band, both $ZnO_2$@NaA and ZnO@NaA shift towards higher frequencies, indicating stress in the zeolite structure induced by the particles [83]. This is consistent with other studies showing that particles interacting with the structure can cause such shifts [84]. Significant changes are also observed in the bending motions of the 6-membered rings, suggesting that zinc species nanoparticles are located at these sites, disrupting the lattice vibrations. This finding aligns with X-ray diffraction and FT-IR data. The latter asymmetric stretching bands are sensitive to changes in the silicon-aluminum ratio and their arrangement.

In Raman mode, the signals decrease sharply after ultrasonication with zinc deposition treatments, especially in the case of ZnO and $ZnO_2$, which indicates that the particles are positioning themselves and altering the building units of the network [85]. As mentioned earlier, this can distort the network and cause these changes. $Zn(OH)_2$ does not exhibit significant changes due to the relatively low quantity of zinc ions. However, these ions are sufficient to induce stress in the zeolite framework, preventing negative charges from forming, as adsorption/desorption is associated with exchangeable ions [86].

The comparison of the thermogravimetric analysis's profiles (Fig 6A) to zeolite NaA and the ultrasonic-treated samples indicates that all samples maintain structural integrity up to 800 °C, despite initial weight loss due to water desorption. Type NaA zeolites exhibit two main regions of mass loss, one near 130 °C and another around 200 °C, which are commonly associated with the loss of physisorbed water molecules from the surface and internal pores of

the zeolite framework [87]. Although these materials exhibit early weight loss, no significant decomposition or framework collapse is observed up to 800 °C, suggesting high thermal robustness. For clarity, thermal stability in this context refers to the absence of structural degradation, as determined by the decomposition onset temperature in DTG curves.

This change was notably observed in the 80–190 ºC range, which we can attribute to the loss of water molecules. Additionally, the second stage of weight loss is reported to be a dihydroxylation process [88], primarily driven by the presence of cations, as they can polarize water molecules [89]. The second phase (Fig 6B) occurs from 280 ºC to 440 ºC, where $ZnO_2$ and ZnO samples are observed. This can be attributed to the particle size within the structure, as it may block the release of water from the zeolite channel system [89,90]. All materials exhibit similar mass loss profiles associated with the aforementioned phenomena between 90 ºC and 450 ºC. Interestingly, a slight increase in final mass was observed in the nanoparticle-containing samples, particularly $ZnO_2$@NaA. This behavior may be attributed to oxidative processes increasing the observed final weight [91].

The zeta potential ($\zeta$) of NaA zeolite shows a negative surface charge (−47 mV), likely due to the formation of Si-O⁻ groups on the surface, which increases the overall negative charge. For $Zn(OH)_2$@NaA and $ZnO_2$@NaA, a further shift to more negative $\zeta$ values is observed (Fig 6A, blue and green lines). This shift can be attributed to ultrasound-induced agglomeration of particles on the zeolite surface, which enhances the exposure of negatively charged Si-O⁻ groups [92]. Interestingly, ZnO@NaA shows a shift toward more positive $\zeta$ values (Fig 6A, red line), which may result from the alignment of $Zn^{2+}$ cations in the [54] plane during the synthesis process. This alignment likely alters the surface charge due to interactions with Si-OH groups, which are exposed and reorganized during the deposition process [93].

The bimodal distribution observed in NaA (Fig 6A, gray line) indicates the presence of two exposed zeolite facets. One corresponds to an exchange site on the [54], which becomes quenched after ion exchange, while the other likely corresponds to the [94], which remains unaffected and has a near-neutral surface charge. For the ZnO species deposited on NaA, the $\zeta$ potential data suggests that the [54] facet is fully covered by ZnO particles, leading to a complete quenching of the signal.

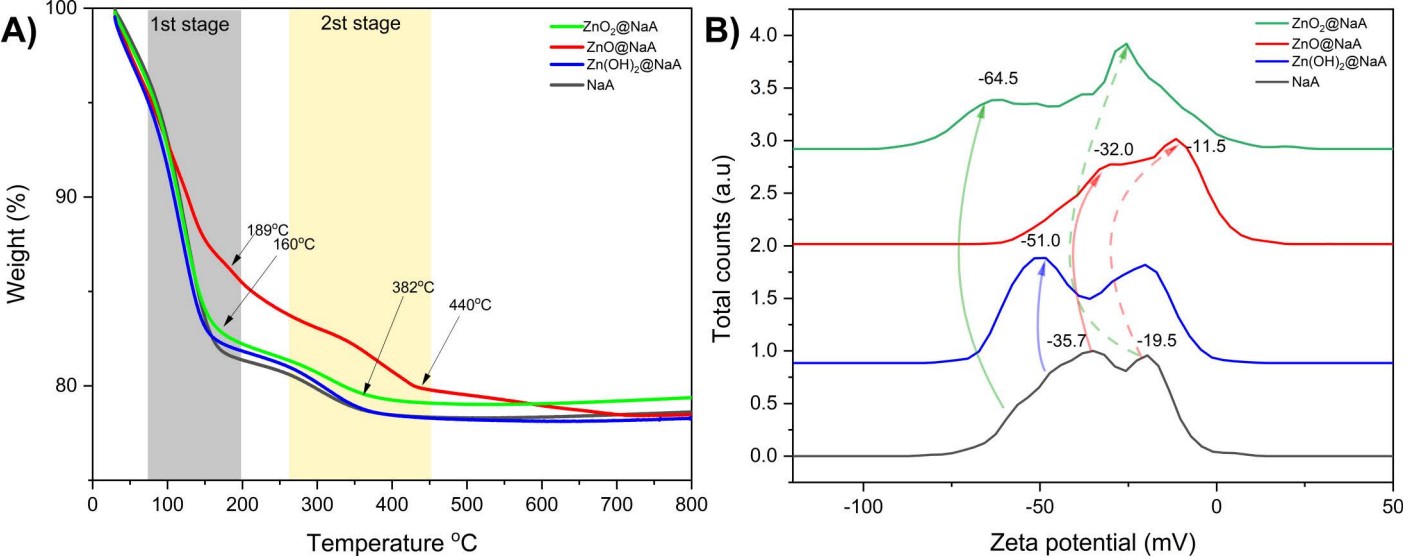

**Fig 6. Zeta potential ($\zeta$) measurements.** A) and thermogravimetric analysis (TGA) of NaA B) modified with $Zn(OH)_2$ (blue line), ZnO (red line), or $ZnO_2$ (green line).

In contrast, for the ZnO$_2$@NaA sample, the [54] facet signal is partially diminished, suggesting that ZnO$_2$ particles cover only a portion of this area, possibly due to particle size and distribution differences. For Zn(OH)$_2$@NaA, no significant change in ζ potential is observed, which aligns with XRD data indicating either a low concentration of Zn(OH)$_2$ or the presence of crystals too small to affect the overall surface charge significantly. This suggests that Zn(OH)$_2$ species are likely located on the inner surface of the zeolite or do not contribute to the external surface charge.

However, a slight shift towards negative values is observed, which could be related to the possible agglomeration of some clusters on the corresponding zeolite facet. This agglomeration could lead to a redistribution of the surface charges of the zeolite. Another potential cause could be increased surface species of metal oxides in zeolites. Therefore, the shifts observed in the samples can be attributed to the presence of zinc species [95].

To further analyze the optical properties of the samples, diffuse reflectance UV-Vis spectroscopy was used to detect the potential Na species [96] and the contribution of the Zn species [97] present in zeolite LTA [94]. The spectral decomposition of the observed bands was used to model the zeolite profile (Fig 7A) and, by subtraction, identify the contribution of the interacting Zn species (Fig 7B–D). The position of the peak max was estimated by employing the second derivative, and previously reported theoretical calculations were used to describe the behavior of zeolite LTA [98]. The detected band energies (eV) were assigned according to intraband and interband transitions determined from the density of states (DOS). Resulting in 11 transitions that correspond to the energy potentials inside the α- and β-cages of phototransferred electrons. The first five potentials in the UV region (199, 215, 234, 260, and 290 nm; 6.2–4.3 eV) are attributed to electronic interactions between confined H$_2$O molecules and Na$^+$ ions forming solvated stabilized ionic pseudoaggregates in the β-cage [96]. These species (called species W) are not yet fully elucidated nor characterized. However, (H$_2$O)$_n$ clusters have been described to be responsible for local electronic redistribution and for the emergence of new excited states of porous materials [99,100]. Through molecular dynamic simulations [99] it was demonstrated that the solvation of protons and the reorganization of H$_2$O in the framework cavities of a zeolite generate electronically active environments. Environments that can stabilize (H$_2$O)$_n$·H clusters when confined in the zeolitic cages (Vener et al.), favoring the formation of extended excitation states [99,100].

As for the bands at 330 nm (3.7 eV) and 446 nm (2.8 eV), these correspond to diamagnetic Na$_8^{6+}$ clusters localized in the β-cage with $A_{1g} \rightarrow T_{1u}$ electronic transitions (associated with the band gap). Equally, the remaining four bands (760–1181 nm) are also optical transitions of Na species in the spherical quantum well with infinite depth and a diameter (d) corresponding to that of the α-cage (d = 1.10–1.15 nm). The band at 763 nm (1.6 eV) is attributed to a surface plasmon-like collective excitation of multiple electrons in Na clusters in the α-cage. The last three bands correspond to Na$_4^{3+}$ clusters, also in the α-cage, being those at 983 nm (1.3 eV) A1g→A2u and A1g→Eu transitions, and at 1079 nm (1.3 eV) and 1181 nm (1.1 eV) single type electron-hole excitations from the 1s→1p orbitals [101]. These mentioned clusters are known to be photoinduced and stabilized due to electron-phonon (vibration/displacement of Na$^+$ ions) interactions and are interpreted as mobile in the α- and β-cages where H$_2$O is absent. This interpretation has been predicted by molecular dynamic simulations and further confirmed by the density of states (DOS), which states the existence of collective excitations in the absence of metallic species [102,103]. Demonstrating that the dynamic configurations of Na$^+$·and H$_2$O in zeolites induced distinctive optical properties [103]. Equally, these interactions contribute to excited states at the visible región, especially when multiple ions are confined in the cavities [102].

The contributions of the Zn species detected from the UV-Vis spectroscopy were assigned by subtracting the red profile that corresponds to NaA (Fig 7B). For the Zn(OH)$_2$@NaA sample, the remainder was attributed to the Zn(OH)$_2$ species that match the characteristic bands with peaks at 290 nm and 380 nm that correspond to Zn(OH)$_2$ [104]. After subtracting the NaA spectrum from the ZnO@NaA spectrum (Fig 7C), two bands, 290 nm and another at 190 nm were observed that can be attributed to ZnO, confirming the presence of these species in the NaA structure [54,105].

Moreover, the decomposition of zeolite NaA absorptions in all samples showed changes in intensity attributed to Zn species interacting strongly with the parent NaA [54,106]. ZnO$_2$@NaA (Fig 7D) shows a sum of bands very similar to

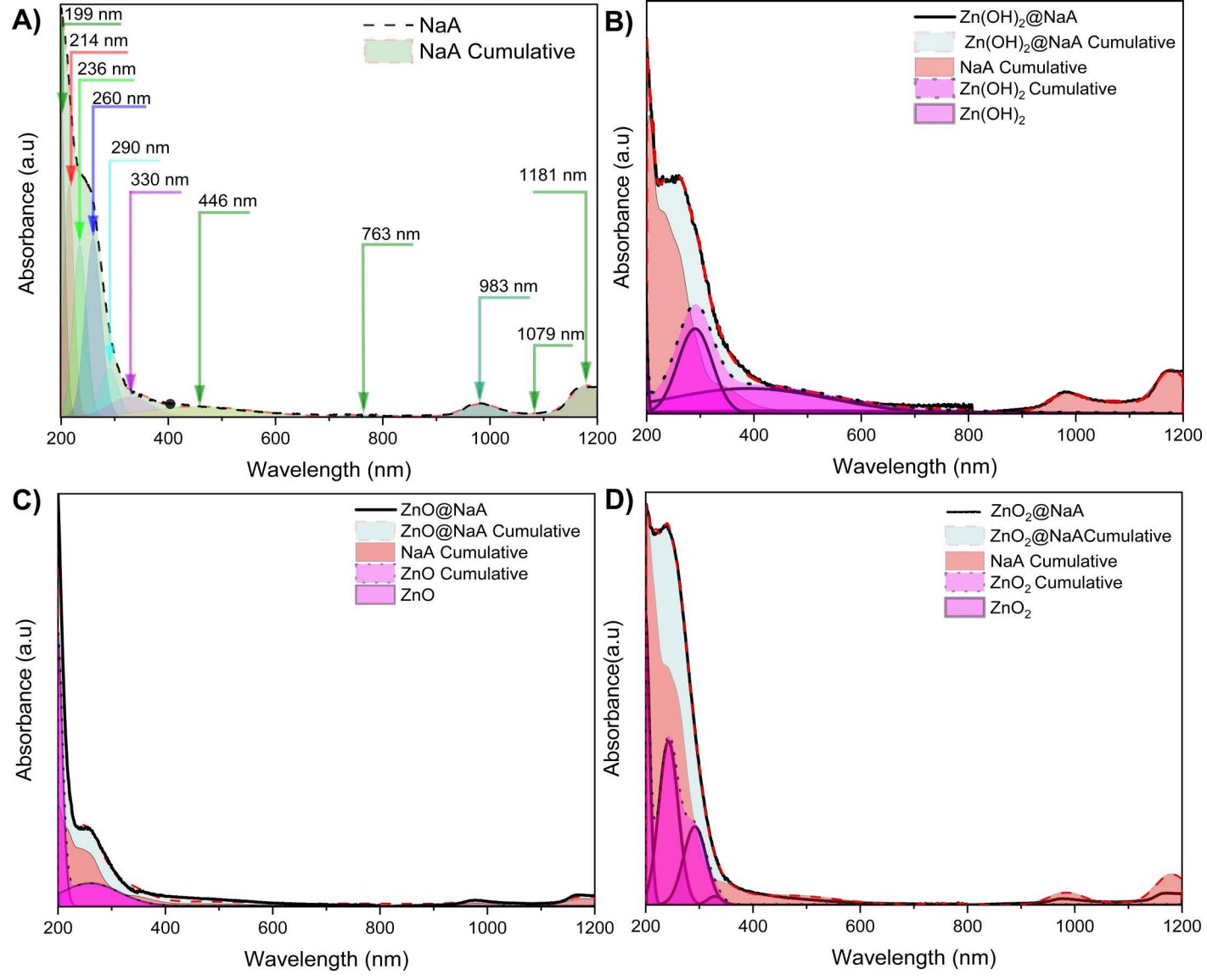

**Fig 7. UV-Vis diffuse reflectance spectroscopy deconvoluted spectra.** A) Deconvolution of zeolite NaA, displaying 11 bands that fit the transition state image published by Roberto Nuñez et al. (2023). B) Bands corresponding to NaA (red) and bands corresponding to zinc species (Pink) $Zn(OH)_2$, C) ZnO, and D) $ZnO_2$.

spectra previously obtained using the same ultrasound method with $ZnO_2$ nanoparticles [38]. The band gaps of each sample show a shift; the calculated band gap for the zeolite was 3.8, while the band gaps for the samples were 3.6 for the $Zn(OH)_2$ and ZnO species and 3.5 for $ZnO_2$. A decrease is observed after the applied treatments. This decrease can be attributed to quantum size effects and band recombination of the materials, possibly reducing the distance between the valence and conduction bands [107]. As can be seen, the samples present a band gap close to 3 eV, indicating the ultraviolet wavelength range, which is crucial for photocatalysis processes [108].

These nanomaterial synthesis methods were compared and evaluated for their efficiency in forming chemical species and deposition on LTA zeolites (Table 4). The analyzed methods include conventional and sono-assisted techniques,

both with and without LTA as a support. Significant differences were observed among the methodologies, particularly in synthesis time, particle morphology, and final material quality. Highlighting the advantages of sono-assisted approaches. In conventional methods without LTA, species such as $Zn(OH)_2$, ZnO, and $ZnO_2$ were formed under acidic conditions and prolonged synthesis times. These methods produced particles with a wide range of sizes and morphologies; however, a tendency toward agglomeration was also detected, especially in the synthesis of $ZnO_2$. Although conventional methods are widely used, their primary limitation is the time-consuming methodologies required to obtain the desired products, which may be inefficient for applications demanding homogeneous and well-dispersed materials.

The use of LTA zeolites as support in conventional methods demonstrated an improvement in the stability of the nanomaterials, though it did not completely resolve the agglomeration issue. For instance, the synthesis of ZnO through conventional ion exchange followed by 5 h of calcination resulted in a relatively uniform material. Still, the prolonged synthesis time and the need for atmospheric control represent significant limitations. Additionally, forming $Ag_2O$ nanoparticles via a similar approach showed comparable challenges, with prolonged synthesis times potentially affecting production efficiency.

In contrast, sono-assisted methods showed a clear advantage by significantly reducing synthesis times and improving the quality of the nanomaterials. US-P enabled the synthesis of $Zn(OH)_2$ in just 15 minutes, while the synthesis of ZnO via ultrasound, using a controlled initial pH solution at 65 °C, improved particle size uniformity. Furthermore, the formation of zinc peroxide through this method occurred within short reaction times, although slight particle agglomeration was observed.

The positive impact of ultrasound was amplified when LTA zeolites were used as a support (this work). Combining sono-assisted methods with LTA reduced reaction times, improved the dispersion of nanomaterials, and increased the surface area of the zeolite. This is particularly relevant for catalytic and adsorption applications, where these properties are essential. Moreover, the combined synthesis approach enabled the efficient and reproducible production of $Zn(OH)_2$, ZnO, and $ZnO_2$.

**Table 4. Comparison of conventional and sono-assisted zinc-based nanomaterial synthesis and deposition methods on LTA.**

| Material | Method | Conclusions | Ref. |
|---|---|---|---|
| $Zn(OH)_2$ | Precipitation | Many particle morphologies in acidic conditions. | [109] |
| ZnO | Precipitation | Time-consuming synthesis and broad particle size. | [109] |
| $ZnO_2$ | Sol-gel | Time-consuming synthesis and NP aggregation. | [110] |
| $Zn(OH)_2$ | US-P | Short synthesis time (15 minutes). | [37] |
| ZnO | US-P | Short synthesis time and NP agglomeration. | [111] |
| $ZnO_2$ | US-SG | Short synthesis time and NP agglomeration. | [38] |
| ZnO@NaA | IMP | Time-consuming synthesis with NP aggregation. | [31] |
| $Zn(OH)_2$-NaA | IE | Time-consuming synthesis with NP aggregation. | [3] |
| $Zn(OH)_2$-NaA | IE | Not characterized. | [25] |
| ZnO@NaA | IE-TO | Time-consuming synthesis with cluster agglomeration. | [25] |
| ZnO@NaA | IE-TO | Time-consuming synthesis with cluster agglomeration. | [24] |
| ZnO@NaA | IE-TO | Time-consuming synthesis of NP. | [91] |
| ZnO@NaA | IE-TO | Time-consuming synthesis with cluster agglomeration. | [112] |
| $Zn(OH)_2$@NaA | US-P | Short synthesis time of Zn species and increased dispersion on the enhanced LTA zeolite surface area. | This Work |
| ZnO@NaA | US-P | | |
| $ZnO_2$@NaA | US-SG | | |

US-P = ultrasound assisted precipitation, US-SG = ultrasound assisted sol-gel, IMP = Impregnation, IE = Ion-Exchange, TO=thermic Oxidation, NP = nanoparticle.

Additionally, when comparing the overall efficiency of the sono-assisted methods to traditional approaches, it becomes evident that sonochemistry offers a more energy-conscious alternative. The use of ultrasonic energy enables rapid nucleation and controlled growth of particles at moderate temperatures (65 °C) and short synthesis times (≤30 min), avoiding prolonged heating cycles required in conventional methods, such as calcination (typically >400 °C for several hours) or hydrothermal synthesis (>100 °C for >12 h) [109,112]. These characteristics significantly reduce the energy input per unit of material obtained, especially when integrated with in-situ deposition strategies.

In addition, the improved material performance, expressed through increased metal dispersion, narrower particle size distribution, and larger surface area, justifies the moderate power demand of ultrasonic equipment. This aligns with our previous studies where sonochemical synthesis is used as an ecological and rapid approach to species formation into zeolites [39,95,113,114].

## Conclusions

The synthesized LTA zeolite in its sodium form, with a Si/Al ratio of 1:1, was successfully modified with Zn species via an ultrasonic treatment. This sono-assisted synthesis and support of $Zn(OH)_2$, ZnO, and $ZnO_2$ nanoparticles, under each specific condition, resulted in a high dispersion within the NaA zeolite, yielding three distinct surface modifications, $Zn(OH)_2$@NaA, ZnO@NaA, and $ZnO_2$@NaA, respectively. The applied ultrasonic treatment did not impact the morphology of the zeolite, nor did it compromise the crystallinity of the LTA structure. However, for sample $Zn(OH)_2$@NaA, the changes were not significant.

In contrast, in samples ZnO@NaA and $ZnO_2$@NaA, from the refinement data, the introduced Zn species were localized at cation exchange site I and in the α-cage, causing a cell contraction from 24.6 Å to 24.4 Å. These results correlated with the values obtained by ICP-OES and EDS, detecting $Zn^{2+}$ in ~50% of cation exchange sites. Where the introduced ZnO or $ZnO_2$ species significantly enhanced the surface area (>400 $m^2$ $g^{-1}$) and pore volume (>0.200 $cm^3$ $g^{-1}$).

These findings suggest that the interaction of Zn species is not merely superficial but is involved in modifying the pore network of the zeolite, effectively depositing the desired zinc species within the NaA structure. This is exhibited by the distortions of NaA when modified with the different Zn species nanoparticles. Given that these distortions were not detected when sonicating without Zn species precursors, the changes in the textural and optical properties of the zeolitic phase were attributed to the sono-assisted Zn modification.

Further studies are needed to increase the concentration of the desired Zn species in a specific local environment and to overcome the challenging selective synthesis of $Zn(OH)_2$ or $ZnO_2$ nanoparticles without obtaining a mixture with the competing ZnO phase. Thus, this applied ultrasonic-assisted methodology holds potential for synthesizing and supporting different types of nanoparticles on zeolites for various applications. Given that synergistic effects occur when combining zeolites with nanoparticles, which enhances the functional properties of the resulting materials. Future research is focused on optimizing the synthesis reaction by considering ultrasonic power, temperature, and time, introducing new functional sites that can enhance catalytic activity and potentially confer antimicrobial properties. These modifications broaden the applicability of the zeolite, making it a promising material for both biomedical and catalytic applications.

## Supporting information

**S1 Table. Pore models and parameters.** Statistical parameters for the obtained models from the $N_2$ adsorption-desorption isotherms.
(DOCX)

**S1 Fig. Speciation diagrams.** A) Speciation of Zn at a concentration of 10 μM at pH = 8 for obtaining $Zn(OH)_2$ and B) 0.10 mM at pH 10 for ZnO.
(DOCX)

**S2 Fig. X-ray diffraction measurements.** Difractograms of Zeolite NaA treated at the ultrasonic conditions to generate $ZnO_2$, $(O_2)$@NaA, and $Zn(OH)_2$, $(OH)$@NaA, without the addition of Zn precursors.
(DOCX)

**S3 Fig. $N_2$ Adsorption-desorption isotherms.** Replicates for the isotherms of zeolite NaA after excessive washing and drying.
(DOCX)

**S1 Dataset. Raw data for X-ray diffraction and $N_2$ Physisorption Isotherms.** Data used for the calculated parameters and calculations of the used samples.
(XLSX)

## Acknowledgments

Jesús Isaías De León Ramírez and Víctor Alfredo Reyes Villegas thank SECIHTI for the doctoral scholarships. We acknowledge Omar Novelo and Ignacio Alejandro Figueroa (IIM-UNAM) for supporting the scanning electron microscopy characterization and Luis Alberto Moreno for Raman spectra measurements. We extend our heartfelt gratitude to Dr. Cesar Diaz✝ for his unwavering support throughout this research.

## Author contributions

**Conceptualization:** Víctor Alfredo Reyes Villegas, Sergio Pérez-Sicairos, Fernando Chávez-Rivas, Vitalii Petranovskii.

**Data curation:** Jesús Isaías De León-Ramírez, Víctor Alfredo Reyes Villegas, José Román Chávez Méndez.

**Formal analysis:** Jesús Isaías De León-Ramírez, Víctor Alfredo Reyes Villegas, Rosario Isidro Yocupicio-Gaxiola.

**Funding acquisition:** José Román Chávez Méndez.

**Investigation:** Jesús Isaías De León-Ramírez, Víctor Alfredo Reyes Villegas, Sergio Pérez-Sicairos, Fernando Chávez-Rivas, Rosario Isidro Yocupicio-Gaxiola.

**Methodology:** Jesús Isaías De León-Ramírez, Víctor Alfredo Reyes Villegas, Rosario Isidro Yocupicio-Gaxiola.

**Project administration:** Jesús Isaías De León-Ramírez, Vitalii Petranovskii.

**Resources:** Jesús Isaías De León-Ramírez, Sergio Pérez-Sicairos, José Román Chávez Méndez, Vitalii Petranovskii.

**Software:** Jesús Isaías De León-Ramírez, Sergio Pérez-Sicairos.

**Supervision:** Jesús Isaías De León-Ramírez, Sergio Pérez-Sicairos, Fernando Chávez-Rivas, Vitalii Petranovskii.

**Writing – original draft:** Jesús Isaías De León-Ramírez, Víctor Alfredo Reyes Villegas.

**Writing – review & editing:** Jesús Isaías De León-Ramírez, Víctor Alfredo Reyes Villegas, Vitalii Petranovskii.

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
