## [Decision Letter · Decision Letter 0]

PONE-D-25-12190Selective Sonochemical post-synthesis Modification of LTA Zeolite with zinc speciesPLOS ONE

Dear Dr. De Leon Ramirez,

Thank you for submitting your manuscript to PLOS ONE. After careful consideration, we feel that it has merit but does not fully meet PLOS ONE’s publication criteria as it currently stands. Therefore, we invite you to submit a revised version of the manuscript that addresses the points raised during the review process.

We look forward to receiving your revised manuscript.

Kind regards,

Mashallah Rezakazemi

Academic Editor

PLOS ONE

Journal Requirements:

“This work was supported by the DGAPA-PAPIIT-UNAM IG101623 Project”

3. Please expand the acronym “UNAM” (as indicated in your financial disclosure) so that it states the name of your funders in full.

“This work was supported by the DGAPA-PAPIIT-UNAM IG101623 Project. J”

“This work was supported by the DGAPA-PAPIIT-UNAM IG101623 Project”

5. We notice that your supplementary figures are included in the manuscript file. Please remove them and upload them with the file type 'Supporting Information'. Please ensure that each Supporting Information file has a legend listed in the manuscript after the references list.

Reviewers' comments:

Reviewer's Responses to Questions

**Comments to the Author**

1. Is the manuscript technically sound, and do the data support the conclusions?

Reviewer #1: Yes

Reviewer #2: Yes

Reviewer #3: Yes

Reviewer #4: Partly

Reviewer #5: Yes

2. Has the statistical analysis been performed appropriately and rigorously? 

Reviewer #1: Yes

Reviewer #2: N/A

Reviewer #3: No

Reviewer #4: N/A

Reviewer #5: Yes

3. Have the authors made all data underlying the findings in their manuscript fully available?

Reviewer #1: Yes

Reviewer #2: Yes

Reviewer #3: Yes

Reviewer #4: No

Reviewer #5: Yes

4. Is the manuscript presented in an intelligible fashion and written in standard English?

Reviewer #1: Yes

Reviewer #2: Yes

Reviewer #3: Yes

Reviewer #4: No

Reviewer #5: Yes

5. Review Comments to the Author

Reviewer #1: 1. Highlighting the roles of the species introduced into the NaA zeolite would be beneficial as to tie up all the characterization techniques that has been employed in this study. Plus the role of sonochemical treatment need to be shown that will affect the modification of properties for NaA zeolite. This can be included in the conclusion part.

2. Since this study emphasize a lot in the role of Zn species in the NaA zeolite, suggestion is to add lattice parameter data for all the exchangeable zeolites to show the significance.

3. The above highlights could be use to further enhance the conclusions with regards to the final morphology of the sonochemically ion exchanges NaA zeolites.

Reviewer #2: A comprehensive and organized investigation of the post-synthesis alteration of NaA (LTA) zeolite with zinc species utilizing a sono-assisted deposition approach is presented in the publication. The study is current and creative, providing an environmentally responsible method of customizing zeolitic materials for cutting-edge uses..

One of the key strengths of this work is the clear and systematic presentation of results. The data is well-supported with relevant references, and the findings are consistently correlated, reinforcing the reliability of the study. The discussion effectively highlights the impact of zinc species incorporation on the physicochemical properties of the modified zeolite, particularly in terms of surface charge redistribution and pore architecture.

Overall, this is a well-executed and insightful piece of work that contributes to the advancement of zeolite modification strategies. The clarity of the methodology, strong correlation between results and discussion, and the practical implications of the study make this manuscript a commendable contribution to the field.

Reviewer #3: the research topic is very interesting and could bring benefit to the scientific community. the research framework focus on the incorporation of various zinc compounds in the zeolites matrices and improving their properties. the introduction points out all the factors necessary for proper understanding of the research. the experimental part give enough information for reader to repeat synthesis and analysis of the zeolites. the results obtained are properly analyzed and discussed. the conclusion summariye all the major findings

the number of figure and tables are satisfactory and give additional value to the article making it more understanable. what NaA means?(please describe it )what was error of determination in Tables 2? what were the major benefits of the Zn incorporation comparing to the LTA?

Reviewer #4: Peer Review PONE-D-25-12190

The manuscript entitled “Selective Sonochemical Post-Synthesis Modification of LTA Zeolite with Zinc Species” employs a sonochemical approach to modify LTA zeolite by depositing zinc-based nanoparticles.

In my opinion, this work is interesting, but I suggest major revisions before further consideration.

GENERAL OBSERVATIONS: The manuscript presents an interesting study; however, it contains numerous typos, grammatical errors, and lacks clarity regarding the experimental conditions. I strongly recommend a thorough review of the text.

Additionally, it remains unclear what type of interaction exists between the zeolite and zinc species (ion exchange, surface deposition, deposition within cavities, etc.). Furthermore, the practical applicability of this composite material is not well elucidated.

Abstract

• The sentence: “In this study, we employed an eco-friendly post-synthesis modification of a NaA zeolite with zinc species: ZnO, ZnO₂, and Zn(OH)₂ by a sono-assisted deposition method. This approach significantly reduced synthesis time.”

o The meaning of "synthesis time" is unclear; please clarify what is being referred to.

• Provide additional information on the interaction between LTA zeolite and Zn species.

Introduction

• Lines 81-87: Review these sentences for clarity and grammatical correctness.

Experimental

• Line 94: Specify the precursor used.

• Line 100: Correct “sonoassisted” to “sono-assisted” if necessary.

• Properly comment on Figure A1 before describing modification methods.

• Line 116: “Adjusting the final pH (according to each Zn species) and volume to 50 mL” — clarify this procedure.

• The passage: “To confirm the effect of the preparation conditions on the zeolite structure, blank samples (excluding the addition of Zn²⁺ precursor solution) were prepared at the deposition solutions (pH, ultrasound duration, and temperature) and studied by X-ray diffraction analysis. Given that no difference was observed, these samples were excluded from the study, presenting the data as an appendix (Figure A2).”

o This statement is unclear; please rephrase for better readability.

• The section on microscopy techniques: “Transmission electron microscopy (TEM) measurements (Hitachi H7500, Hitachi Ltd., Tokyo, Japan) at 80 kV. Sample morphology was obtained by scanning electron microscopy (SEM) using a JEOL JIB-4500 (MA, USA). The images were processed in Image J imaging software, and a second set of samples was analyzed on a LEICA Stereoscan 440 Scanning Electron Microscope. The samples were pretreated with a gold sputtering coating. The energy dispersive X-ray spectroscopy (EDX) coupled to the microscope used for SEM was also recorded.”

o This section requires revision; specify the EDX instrument used and clarify why two different SEM instruments were employed for two different sets of samples. Which samples were analyzed by each method? The same applies to ICP-OES.

• “A quantity between 25-50 mg was dissolved…” — clarify what was dissolved.

• “The samples were degassed at 50°C for 60 minutes, followed by a heating ramp with a rate of 5°C min⁻¹ until reaching 250°C, maintaining this temperature for 360 minutes before the analysis.”

o Explain why this thermal treatment was performed.

• “The study was conducted from 30°C to 800°C, with a heating ramp of 20°C min⁻¹ in an O₂ atmosphere.”

o Specify whether the atmosphere was pure O₂ or a mixture (e.g., N₂/O₂).

• Provide additional details regarding the experimental conditions for FTIR and Raman spectroscopy.

• Figure 3: The authors present N₂ sorption/desorption isotherms, but this method is not described in the experimental section. A proper description should be included.

Results and Discussion

• The term “thermal stability” is unclear. The samples appear to be thermally unstable, as they exhibit significant weight loss at low temperatures. Thermal stability (or oxidative stability in this case) should be defined based on onset temperature from DTG curves.

• Sample ZnO₂@NaA exhibits a weight gain at higher temperatures, which may be due to nitrogen adsorption from the furnace atmosphere, as also observed in isotherm measurements.

• Figure 6: It is recommended to invert the order of A and B to align with the discussion in the text.

Overall, the manuscript requires substantial revision for clarity, grammar, and experimental detail. The authors should ensure that all methods are thoroughly described and justified, particularly regarding sample preparation and characterization techniques.

Reviewer #5: Dear Editor,

I have carefully read the manuscript PONE-D-25-12190, entitled "Selective Sonochemical Post-Synthesis Modification of LTA Zeolite with Zinc Species", by J. I. De León-Ramírez et al., submitted to PLOS ONE.

Overall, I found this manuscript interesting, and I believe it merits publication in PLOS ONE after revision.

- First, the entire document should be reviewed by a native English speaker. The structure of some paragraphs is clearly inadequate but can be easily corrected. This would improve the readability and coherence of the text.

- The authors should improve the flow of the text around line 67. There is a noticeable gap that disrupts the reading at this point.

- In line 76, the authors state: "More recently, sonochemical approaches, such as sono-assisted precipitation (P-US), have emerged as innovative techniques for modifying zeolites." I believe this topic could be further developed. As it stands, the introduction section is weak. The authors should expand the literature review for each topic covered and include more details about the current state of the art. Based on this, clearly define the research gap and explain the limitations of previous studies in relation to your contribution. What is your novelty, specifically? While it becomes apparent throughout the manuscript, it needs to be stated more directly in the introduction to convince the reader.

- The methodology is clear and provides sufficient detail for reproducing the research independently—an essential aspect of scientific work. However, I encourage the authors to justify each methodological choice with appropriate citations.

- Figure 1(a): the graph presented alongside the image is unclear. While the authors successfully conveyed their message in parts (b)–(d), the same cannot be said for part (a). Please revise and improve its clarity.

- As a general comment, I would appreciate a comparison of the efficiency of the material synthesized in this work with that of equivalent materials produced using other techniques. An analysis of energy demand versus efficiency in the context of the sonochemical treatment would greatly enhance the value of this work.

- The conclusion section should be more quantitative. Include specific data to strengthen this section.

An excellent discussion on the characterization of the synthesized material was provided, which adds significant value to this manuscript. In my opinion, the manuscript can be considered for publication in PLOS ONE once these points are carefully addressed.

6. PLOS authors have the option to publish the peer review history of their article (what does this mean? ). If published, this will include your full peer review and any attached files.

**Do you want your identity to be public for this peer review?** For information about this choice, including consent withdrawal, please see our Privacy Policy .

Reviewer #1: No

Reviewer #2: No

Reviewer #3: **Yes: ** Tomislav Tosti

Reviewer #4: No

Reviewer #5: No

---

## [Author Response · Author response to Decision Letter 1]

22 Apr 2025

Journal Requirements:

Authors Response:

We have reviewed and corrected the format for the titles, images and tables in line with the journals style requirements.

“This work was supported by the DGAPA-PAPIIT-UNAM IG101623 Project”

Authors Response:

The funders role has been specified in the cover letter as suggested.

3. Please expand the acronym “UNAM” (as indicated in your financial disclosure) so that it states the name of your funders in full.

Authors Response:

The acronym UNAM has been expanded stating the name of the funders in full as well as the other funders.

“This work was supported by the DGAPA-PAPIIT-UNAM IG101623 Project. J”

“This work was supported by the DGAPA-PAPIIT-UNAM IG101623 Project”

Authors Response:

The financial disclosure has been removed fromed from the Acknowledgments section of the manuscript and added to the cover letter for further inclusion in the Funding Statement section.

5. We notice that your supplementary figures are included in the manuscript file. Please remove them and upload them with the file type 'Supporting Information'. Please ensure that each Supporting Information file has a legend listed in the manuscript after the references list.

Authors Response:

All supplementary figures have been stated in the text as required by the style format. Equally, they have been removed from the manuscript, adding all these figures in a separate file.

Reviewers' comments:

Reviewer's Responses to Questions

Comments to the Author

1. Is the manuscript technically sound, and do the data support the conclusions?

Reviewer #1: Yes

Reviewer #2: Yes

Reviewer #3: Yes

Reviewer #4: Partly

Reviewer #5: Yes

Authors Response:

We thank all reviewers for their time and comments. We are pleased that Reviewers #1, #2, #3, and #5 fully supported the manuscript. We appreciate Reviewer #4’s partial recommendation and have addressed their concerns carefully. Revisions have been made to improve clarity, methodological detail, and data interpretation. We believe the updated manuscript now meets the journal’s standards for technical rigor and scientific validity.

2. Has the statistical analysis been performed appropriately and rigorously?

Reviewer #1: Yes

Reviewer #2: N/A

Reviewer #3: No

Reviewer #4: N/A

Reviewer #5: Yes

Authors Response:

We appreciate the reviewer’s concern regarding the statistical analysis. In response, we have revised the relevant sections to clarify the statistical methods used, including the tests applied, reporting p-values and confidence intervals where appropriate. All analyses were reviewed to ensure they were rigorous and appropriate for the data presented. These updates are highlighted in the revised manuscript.

3. Have the authors made all data underlying the findings in their manuscript fully available?

Reviewer #1: Yes

Reviewer #2: Yes

Reviewer #3: Yes

Reviewer #4: No

Reviewer #5: Yes

Authors Response:

We thank the Reviewers for confirming that the data underlying our findings have been made fully available. As per PLOS Data Policy, all relevant data are provided within the manuscript and its Supporting Information files. We remain committed to transparency and reproducibility, and we are glad the reviewers found our data sharing practices appropriate. However, we have also stated in the “Data Availability Statement” that data can be obtained upon reasonable request.________________________________________

4. Is the manuscript presented in an intelligible fashion and written in standard English?

Reviewer #1: Yes

Reviewer #2: Yes

Reviewer #3: Yes

Reviewer #4: No

Reviewer #5: Yes

Authors Response:

We thank the Reviewers for confirming the clarity and language quality of the manuscript. Nevertheless, according to all the suggestions from these, we have thoroughly revised the text to correct minor grammatical and typographical issues and to improve overall clarity. The revised manuscript has been carefully edited for standard English usage by native English speakers and Grammarly, to ensure it is clear and unambiguous throughout.________________________________________

5. Review Comments to the Author

Reviewer #1: 1. Highlighting the roles of the species introduced into the NaA zeolite would be beneficial as to tie up all the characterization techniques that has been employed in this study. Plus the role of sonochemical treatment need to be shown that will affect the modification of properties for NaA zeolite. This can be included in the conclusion part.

Authors Response:

We have revised the Conclusion section to explicitly highlight the roles of the introduced species in modifying the NaA zeolite structure and properties, linking these effects to the results from the characterization techniques employed. Additionally, we have emphasized the impact of sonochemical treatment on enhancing dispersion, promoting ion exchange, and modifying surface characteristics. These additions help to better integrate the findings and clarify the contributions of both the introduced species and the sonochemical process.

Changes in text

The synthesized LTA zeolite in its sodium form, with a Si/Al ratio of 1:1, was successfully modified with Zn species via an ultrasonic treatment. TheThis sono-assisted synthesis and support of Zn(OH)2, ZnO, and ZnO2 nanoparticles, under each specific conditions, were synthesized and supported withcondition, resulted in a high dispersion within the NaA zeolite, yielding three distinct samplessurface modifications, Zn(OH)2@NaA, ZnO@NaA, and ZnO2@NaA, respectively. The applied ultrasonic treatment did not impact the morphology of the zeolite, effectively depositing the desired zinc species within the NaA structure. However, further studies are needed to increase the concentration of the deposited Zn species. It is challenging to selectively synthesize Zn(OH)2 or ZnO2 nanoparticles without obtaining the competing ZnO phase.nor did it compromise the crystallinity of the LTA structure. However, for sample Zn(OH)2@NaA, the changes were not significant.

TheIn contrast, in samples ZnO@NaA and ZnO₂@NaA, from the refinement data, the introduced Zn-modified zeolites species were localized at cation exchange site I and in the α-cage, causing a cell contraction from 24.6 Å to 24.4 Å. These results correlated with the values obtained by ICP-OES and EDS, detecting Zn2+ in ~50% of cation exchange sites. Where the introduced ZnO or ZnO2 species significantly enhanced the surface area (>400 m2 g-1) and pore volume (>0.200 cm3 g-1).

These findings suggest that the interaction of Zn species is not merely superficial but is involved in modifying the pore network of the zeolite, effectively depositing the desired zinc species within the NaA structure. This is exhibited a polycrystalline structure with by the distortions caused by of NaA when modified with the different Zn species nanoparticles. Where the pore size of NaA was significantly affected, increasing both pore size and surface area. Potentially enhancing the adsorptionGiven that these distortions were not detected when sonicating without Zn species precursors, the changes in the textural and optical properties of Zn-modified NaAthe zeolitic phase were attributed to the sono-assisted Zn modification.

Further studies are needed to increase the concentration of the desired Zn species in a specific local environment and to overcome the challenging selective synthesis of Zn(OH)2 or ZnO2 nanoparticles without obtaining a mixture with the competing ZnO phase. Thus, this applied ultrasonic-assisted methodology holds potential for synthesizing and supporting different types of nanoparticles on zeolites for various applications. Given that synergistic effects occur when combining zeolites with nanoparticles, which enhances the functional properties of the resulting materials. Future research is focused on optimizing the synthesis reaction by considering ultrasonic power, temperature, and time, expanding their applicability in different fieldsintroducing new functional sites that can enhance catalytic activity and potentially confer antimicrobial properties. These modifications broaden the applicability of the zeolite, making it a promising material for both biomedical and catalytic applications.

2. Since this study emphasize a lot in the role of Zn species in the NaA zeolite, suggestion is to add lattice parameter data for all the exchangeable zeolites to show the significance.

Authors Response:

We appreciate the reviewer’s insightful suggestion. In response, we have added the lattice parameter data and Rwp from the Retvield refinement for the NaA and Zn-modified zeolites to better illustrate the structural changes. This addition supports the discussion on the role of Zn species and highlights the comparative effects of the different Zn species on the NaA zeolite framework.

Changes in text

Table 3: Displacements of planes relative to NaA, where negative values indicate leftward shifts and positive values indicate rightward shifts.

Table 3. Displacements of planes relative to NaA, where negative values indicate leftward shifts and positive values indicate rightward shifts.

Plane Peak Position (2θ) peak position shift (Δ2θ)*

NaA Zn(OH)2@NaA ZnO@NaA ZnO2@NaA

(200) 7.18 -0.04 0.03 0.04

(220) 10.16 -0.02 0.04 0.06

(222) 12.44 -0.02 0.09 0.11

(420) 16.09 -0.02 0.11 0.15

(620) 21.65 -0.02 0.15 0.21

(622) 23.96 -0.02 0.20 0.25

(642) 21.66 -0.04 0.15 0.19

*The values represent the peak position shift between the samples (Sample NaA).a=b=c (Å) 24.6 24.6 24.4 24.4

Rwp 8.5 12.8 8.1 13.1

Rwp = Rp- Rexpected *The values represent the peak shift difference of the samples (Δ2θ=Sample-NaA).

3. The above highlights could be use to further enhance the conclusions with regards to the final morphology of the sonochemically ion exchanges NaA zeolites.

Authors Response:

Thank you for this helpful observation. Following your suggestion, we have revised the Conclusion section to better integrate the structural data (including lattice parameters) and morphological observations. These enhancements emphasize how the nature of the Zn species and the sonochemical treatment collectively influence the final morphology and framework integrity of the NaA zeolites. This addition provides a more comprehensive summary that aligns with the study’s findings and characterization results.

Reviewer #2: A comprehensive and organized investigation of the post-synthesis alteration of NaA (LTA) zeolite with zinc species utilizing a sono-assisted deposition approach is presented in the publication. The study is current and creative, providing an environmentally responsible method of customizing zeolitic materials for cutting-edge uses.

Authors Response:

We sincerely thank the reviewer for their positive and encouraging remarks regarding the novelty, environmental relevance, and organization of our study. We are pleased that the sono-assisted approach and its potential for tailoring zeolitic materials for advanced applications were well received. We have carefully considered all suggestions provided and have revised the manuscript accordingly to further improve its clarity and scientific value.

One of the key strengths of this work is the clear and systematic presentation of results. The data is well-supported with relevant references, and the findings are consistently correlated, reinforcing the reliability of the study. The discussion effectively highlights the impact of zinc species incorporation on the physicochemical properties of the modified zeolite, particularly in terms of surface charge redistribution and pore architecture.

Authors Response:

We greatly appreciate the reviewer’s positive feedback on the clarity and organization of the results, as well as the consistency and reliability of the data interpretation. We are especially grateful for the recognition of our efforts to correlate the incorporation of zinc species with changes in surface charge and pore architecture. This aspect was central to our investigation, and we are pleased that it came across effectively in the discussion.

Overall, this is a well-executed and insightful piece of work that contributes to the advancement of zeolite modification strategies. The clarity of the methodology, strong correlation between results and discussion, and the practical implications of the study make this manuscript a commendable contribution to the field.

Authors Response:

We sincerely thank the reviewer for the kind and encouraging comments. We are pleased that the clarity of the methodology, the strength of the data interpretation, and the practical implications of our findings were appreciated. It is rewarding to know that our work is seen as a valuable contribution to the advancement of zeolite modification strategies.

Reviewer #3: the research topic is very interesting and could bring benefit to the scientific community. the research framework focus on the inc

---

## [Decision Letter · Decision Letter 1]

Selective Sonochemical post-synthesis Modification of LTA Zeolite with zinc species

PONE-D-25-12190R1

Dear Dr. De Leon Ramirez,

We’re pleased to inform you that your manuscript has been judged scientifically suitable for publication and will be formally accepted for publication once it meets all outstanding technical requirements.

Kind regards,

Mashallah Rezakazemi

Academic Editor

PLOS ONE

Additional Editor Comments (optional):

Reviewers' comments:

Reviewer's Responses to Questions

**Comments to the Author**

1. If the authors have adequately addressed your comments raised in a previous round of review and you feel that this manuscript is now acceptable for publication, you may indicate that here to bypass the “Comments to the Author” section, enter your conflict of interest statement in the “Confidential to Editor” section, and submit your "Accept" recommendation.

Reviewer #2: All comments have been addressed

Reviewer #3: All comments have been addressed

Reviewer #5: All comments have been addressed

2. Is the manuscript technically sound, and do the data support the conclusions?

Reviewer #2: (No Response)

Reviewer #3: Yes

Reviewer #5: Yes

3. Has the statistical analysis been performed appropriately and rigorously? 

Reviewer #2: (No Response)

Reviewer #3: Yes

Reviewer #5: Yes

4. Have the authors made all data underlying the findings in their manuscript fully available?

Reviewer #2: (No Response)

Reviewer #3: Yes

Reviewer #5: Yes

5. Is the manuscript presented in an intelligible fashion and written in standard English?

Reviewer #2: (No Response)

Reviewer #3: Yes

Reviewer #5: Yes

6. Review Comments to the Author

Reviewer #2: (No Response)

Reviewer #3: the authors made an effort to fulfill the reviewers' suggestions and improve article quality. the implemented changes brings improvements and article meets minimum requirements for publication. But I have one more remark regarding abbreviations in abstract in my opinion the abstract must contain full name especially for first time mentioning.

Reviewer #5: Tha authors reviewed the manuscript criteriously. My questions were properly answered. I also noticed they did a good job answering the questions from the other reviewers. The manuscript can now be accepted for publication.

7. PLOS authors have the option to publish the peer review history of their article (what does this mean? ). If published, this will include your full peer review and any attached files.

**Do you want your identity to be public for this peer review?** For information about this choice, including consent withdrawal, please see our Privacy Policy .

Reviewer #2: No

Reviewer #3: **Yes: ** Tomislav Tosti

Reviewer #5: No

---

## [Editor Report · Acceptance letter]

PONE-D-25-12190R1

PLOS ONE

Dear Dr. De León-Ramírez,

I'm pleased to inform you that your manuscript has been deemed suitable for publication in PLOS ONE. Congratulations! Your manuscript is now being handed over to our production team.

Kind regards,

on behalf of

Dr. Mashallah Rezakazemi

Academic Editor

PLOS ONE